# WHEN TO USE GRAPHS IN RAG: A COMPREHENSIVE ANALYSIS FOR GRAPH RETRIEVAL-AUGMENTED GENERATION

**Zhishang Xiang**[1,*] **Chuanjie Wu**[2,*], **Qinggang Zhang**[4,†], **Shengyuan Chen**[4], **Zijin Hong**[4]
**Xiao Huang**[4], **Jinsong Su**[2,1,3,†]
[1]Institute of Artificial Intelligence, Xiamen University, China
[2]School of Informatics, Xiamen University, China
[3]Key Laboratory of Digital Protection and Intelligent Processing of Intangible Cultural Heritage
of Fujian and Taiwan (Xiamen University), Ministry of Culture and Tourism, China
[4]The Hong Kong Polytechnic University, Hong Kong SAR, China
```
xiangzhishang@stu.xmu.edu.cn; wuchuanjie@stu.xmu.edu.cn;
qinggangg.zhang@connect.polyu.hk; zijin.hong@connect.polyu.hk;
{sheng-yuan.chen, xiao.huang}@polyu.edu.hk; jssu@xmu.edu.cn
```

## ABSTRACT

Graph retrieval-augmented generation (GraphRAG) has emerged as a powerful paradigm for enhancing large language models (LLMs) with external knowledge. It leverages graphs to model the hierarchical structure between specific concepts, enabling more coherent and effective knowledge retrieval for accurate reasoning. Despite its conceptual promise, recent studies report that GraphRAG frequently underperforms vanilla RAG on many real-world tasks. This raises a critical question: Is GraphRAG really effective, and in which scenarios do graph structures provide measurable benefits for RAG systems? To address this, we propose GraphRAG-Bench, a comprehensive benchmark designed to evaluate GraphRAG models on both hierarchical knowledge retrieval and deep contextual reasoning. GraphRAG-Bench features a comprehensive dataset with tasks of increasing difficulty, covering fact retrieval, complex reasoning, contextual summarize, and creative generation, and a systematic evaluation across the entire pipeline, from graph construction and knowledge retrieval to final generation. Leveraging this novel benchmark, we systematically investigate the conditions when GraphRAG surpasses traditional RAG and the underlying reasons for its success, offering guidelines for its practical application. All related resources and analysis are collected for the community at https://github.com/GraphRAG-Bench/GraphRAG-Benchmark.

## 1 INTRODUCTION

Large language models (LLMs), like Claude (Anthropic, 2024) and GPT (OpenAI, 2023) series, have surprised the world with their remarkable capabilities in many real-world tasks, like linguistic comprehension (Brown et al., 2020), question answering (Khashabi et al., 2020), mathematical reasoning (Hong et al., 2025), and content generation (Chowdhery et al., 2023; Hong et al., 2024; Zhang et al., 2024a). Despite the success, LLMs are always criticized for their inability to handle knowledge-intensive tasks and the tendency to generate hallucinations (Huang et al., 2023), especially when faced with questions requiring specialized expertise (Chen et al., 2024b; He et al., 2024; Tan et al., 2024). Retrieval-augmented generation (RAG) (Gao et al., 2023; Lewis et al., 2020) has recently offered a promising approach to adapt LLMs for specific or private domains. Rather than retraining LLMs to incorporate new knowledge and updates (Feng et al., 2025; Fang et al., 2025; Jiang et al., 2025; Wang et al., 2024b; Zhang et al., 2025b), RAG enhances these models by leveraging external knowledge from text corpora. This approach enables LLMs to generate responses by leveraging not only their parametric knowledge but also real-time retrieved domain-specific information, thereby providing more accurate and reliable answers (Chen et al., 2024a; Li et al., 2024).

---

*Equal contribution. †Corresponding author.

However, traditional RAG systems often face critical challenges when dealing with large-scale, unstructured domain corpora (Edge et al., 2024; Peng et al., 2024). The textual documents in this corpus, collected from different sources, like research papers, textbooks and technical reports, often vary widely in accuracy and completeness (Guo et al., 2025). The information retrieved by RAG systems can be extensive, complex, and lack clear organization, since domain knowledge is typically scattered across multiple documents without clear hierarchical relationships between different concepts (Sun et al., 2024; Zhang et al., 2024b; Ma et al., 2024). Although RAG systems (Borgeaud et al., 2022; Izacard et al., 2023; Jiang et al., 2023) attempt to manage this complexity by dividing documents into smaller chunks for effective indexing, this approach inadvertently sacrifices crucial contextual information, significantly compromising retrieval accuracy and contextual comprehension for complex reasoning (Han et al., 2024; Zhang et al., 2025a; Shengyuan et al., 2024).

To address this, graph retrieval-augmented generation (GraphRAG) (Zhang et al., 2025a; Peng et al., 2024; Procko & Ochoa, 2024) has recently emerged as a powerful paradigm that leverages external structured graphs to improve LLMs' capability on contextual comprehension (Han et al., 2024; Zhang et al., 2025a). Early efforts, like Microsoft GraphRAG (Edge et al., 2024) and its variant LazyGraphRAG (Darren Edge, 2024), employ hierarchical community-based search and combine local/global querying for comprehensive responses. Building on this, LightRAG (Guo et al., 2024) improves scalability through dual-level retrieval and graph-enhanced indexing, while GRAG (Hu et al., 2024) introduces a soft pruning technique to mitigate the impact of irrelevant entities in retrieved subgraphs and employs graph-aware prompt tuning to help LLMs interpret topological structure. Further extending these capabilities, StructRAG (Li et al., 2024) tailors data structures to specific tasks by dynamically selecting optimal graph schemas, while KAG (Liang et al., 2024) constructs domain expert knowledge using conceptual semantic reasoning and human-annotated schemas, which significantly reduces noise present in OpenIE systems. These strategies used in GraphRAG models significantly improve retrieval precision and contextual depth, enabling LLMs to address complex, multi-hop queries more effectively.

Despite its conceptual promise, recent studies (Han et al., 2025; Zhou et al., 2025) report that GraphRAG models frequently underperform traditional RAG approaches on many real-world tasks. Specifically, the previous study (Han et al., 2025) demonstrates that GraphRAG achieves 13.4% lower accuracy on Natural Question compared to vanilla RAG, with particularly poor performance on time-sensitive queries (e.g., 16.6% accuracy drop for questions requiring real-time knowledge updates). While graph retrieval improves reasoning depth by 4.5% on HotpotQA's multi-hop questions, it introduces 2.3 × higher latency on average (Zhou et al., 2025). These inconsistencies between conceptual potential and practical efficacy raise critical questions: **Is GraphRAG really effective, and in which scenarios do graph structures provide measurable benefits for RAG systems?**

It is crucial to identify the factors that are currently limiting GraphRAG's real-world performance. However, quantitatively and fairly assessing the role of graph structures in RAG systems is challenging. Current benchmarks, including HotpotQA (Yang et al., 2018), MultiHopRAG (Tang & Yang, 2024) and UltraDomain (Qian et al., 2024), fail to adequately evaluate the effectiveness of graph structures in RAG systems due to fundamental limitations in both their problem design and corpus composition. ❶ First, existing benchmarks **lack granular differentiation in task complexity**. Existing benchmarks overemphasize retrieval difficulty, locating scattered facts from corpora, while neglecting reasoning complexity, which involves synthesizing interconnected facts into contextually grounded solutions. As shown in Figure 2, they predominantly focus on narrow task categories, such as simple fact retrieval or linear multi-hop reasoning, without systematically capturing the spectrum of challenges encountered in real-world scenarios (Tang & Yang, 2024). For instance, a typical multi-hop question in existing benchmarks might ask, "Who founded Company `Kjaer Weis`, and in which city was this person born¿' This requires only the extraction of several discrete facts and cannot extend to complex scenarios requiring hierarchical reasoning and contextual synthesis. ❷ Second, corpora in existing RAG benchmarks suffer from **inconsistent quality and low information density**. Many datasets are built on generic sources like Wikipedia or news articles, which lack domain-specific knowledge or explicit logical connections. While some work, like UltraDomain (Qian et al., 2024), has tried to extract domain-specific corpora from textbooks, they often fail to encode implicit hierarchies for real-world applications. This makes it impossible to assess GraphRAG's core strengths in leveraging domain hierarchies. For example, a corpus with poorly defined conceptual hierarchies or loosely connected entities cannot meaningfully test whether graph-aware retrieval mechanisms improve multi-hop reasoning or preserve contextual coherence during knowledge acquisition. Additionally,

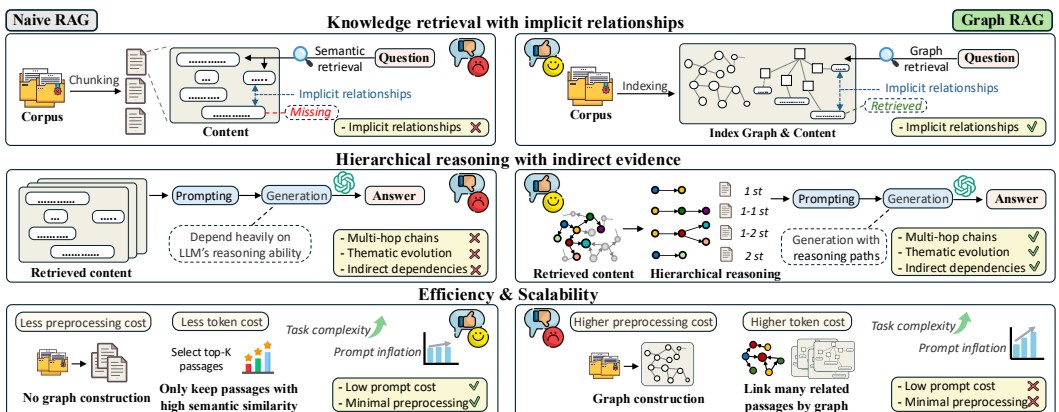

Figure 1: RAG vs. GraphRAG. The pipelines of RAG and GraphRAG and their characteristics.

the absence of corpora with varying information densities, ranging from tightly structured domain knowledge to loosely organized real-world documents, further restricts the evaluation of graph structures' scalability and adaptability.

To bridge this gap, we propose GraphRAG-Bench, a comprehensive benchmark designed to evaluate GraphRAG models on deep reasoning. GraphRAG-Bench features ❶ **comprehensive corpora** with different information density, including tightly structured domain knowledge and loosely organized texts, and ❷ **tasks of increasing difficulty**, covering fact retrieval, multi-hop reasoning, Contextual Summarize, and creative generation, and ❸ **systematic evaluation** across the entire pipeline, from graph construction and knowledge retrieval to final generation. Leveraging this novel benchmark, we systematically investigate the conditions when GraphRAG surpasses traditional RAG systems and the underlying reasons for its success, offering guidelines for its practical application.

## 2 PRELIMINARY STUDY

Before going into the details of our benchmark, we first examine the pipelines of RAG and GraphRAG, and conduct a comprehensive study to identify the primary limitation of existing benchmarks.

### 2.1 RAG VS. GRAPHRAG

We carefully compare GraphRAG's pipeline with traditional RAG's and summarize their characteristics in Figure 1. Generally, RAG retrieves contextually relevant data from a corpus during inference, enabling real-time, domain-specific responses without model retraining. While efficient, its reliance on direct semantic similarity may overlook the broader contextual web of relationships, hierarchies, or implicit logic that binds concepts together. GraphRAG addresses this limitation by expanding the retrieval framework beyond semantic relevance. It structures background knowledge as a graph, where nodes represent entities, events, or themes, and edges define their logical, causal, or associative connections. When processing a query, GraphRAG retrieves not only directly related nodes but also traverses the graph to capture interconnected subgraphs, uncovering latent patterns such as thematic evolution, indirect dependencies, or multi-step reasoning chains. This approach enables the model to synthesize insights from dispersed data points, making it particularly good at tasks demanding complex logical inference. For instance, while RAG might retrieve isolated facts about a topic, GraphRAG could identify related events, causal chains, or thematic clusters, thereby enabling more coherent and comprehensive responses. To sum up, the primary distinction between these two paradigms lies in their handling of contextual depth. RAG excels in scenarios requiring rapid access to discrete information, while GraphRAG emphasizes deep contextual analysis for tasks requiring nuanced understanding of interconnected data. More detailed analysis is included in Appendix I.

### 2.2 CURRENT RAG BENCHMARKS

Existing benchmarks, such as HotpotQA (Yang et al., 2018), MultiHopRAG (Tang & Yang, 2024) and UltraDomain (Qian et al., 2024), were primarily designed to evaluate traditional text-centric

RAG frameworks. While these benchmarks have advanced the field, they exhibit critical limitations when applied to assessing GraphRAG.

First, existing benchmarks narrowly focus on testing `retrieval difficulty`, the ability to locate scattered information from the corpus, while neglecting the equally critical challenge of `reasoning difficulty`, which involves integrating interconnected concepts/facts by capturing the latent logic. While these benchmarks include "multi-hop" questions to test a model's ability in complex reasoning, they do not reflect real-world scenarios demanding complex logical synthesis.

For instance, a typical multi-hop question in existing benchmarks might ask, "Who founded Company `Kjaer Weis`, and in which city was this person born?" This requires only the extraction of several discrete facts from the corpus. However, real-world problems, such as explaining why Company `Kjaer Weis` failed in a specific market, demand synthesizing financial reports, competitor analyses, consumer trends, and regulatory changes into a coherent narrative. GraphRAG's strength lies in mapping these interdependencies (e.g., "Market entry timing → supply chain disruptions → regulatory fines → brand erosion") through graph traversals. However, current benchmarks lack tasks that explicitly require such synthesis, reducing "multi-hop" queries to sequential fact retrieval within narrow contexts, failing to evaluate how models infer domain-specific hierarchies.

Table 1: Categorization of tasks by complexity, ranging from factual retrieval to creative generation.

| Category | Task Name | Brief Description | Example |
|---|---|---|---|
| Level 1 | Fact Retrieval | Require retrieving isolated knowledge points with minimal reasoning; mainly test precise keyword matching. | *Which region of France is Mont St. Michel located?* |
| Level 2 | Complex Reasoning | Require chaining multiple knowledge points across documents via logical connections. | *How did Hinze's agreement with Felicia relate to the perception of England's rulers?* |
| Level 3 | Contextual Summarize | Involve synthesizing fragmented information into a coherent, structured answer; emphasize logical coherence and context. | *What role does John Curgenven play as a Cornish boatman for the visitors exploring this region?* |
| Level 4 | Creative Generation | Require inference beyond retrieved content, often involving hypothetical or novel scenarios. | *Retell the scene of King Arthur's comparison to John Curgenven and the exploration of the Cornish coastline as a newspaper article.* |

Second, the corpora used in existing benchmarks suffer from inconsistent quality and low information density. Most datasets are built on generic sources like Wikipedia or news articles, which lack structured domain-specific knowledge or explicit logical connections. A corpus with poorly defined conceptual hierarchies or loosely connected entities cannot meaningfully test whether graph-aware retrieval mechanisms improve multi-hop reasoning or preserve contextual coherence during knowledge integration. While some work, like UltraDomain, has tried to construct domain-specific corpora using textbooks, they often fail to encode implicit hierarchies for real-world applications. As shown in Table 2 and Table 13 in Appendix E, domain concepts and their hierarchical dependency appear sparsely in the corpus. This sparsity falls far below the threshold of multi-hop reasoning, which makes it impossible to assess GraphRAG's core strengths in leveraging domain hierarchies. Additionally, the absence of corpora with varying information densities, ranging from tightly structured domain knowledge to loosely organized real-world documents, further restricts the evaluation of graph structures' scalability.

Table 2: Average number of entities and relations across benchmarks. (Details are in Table 13 in Appendix E)

| Metric | Ultradomain | MultiHop-RAG | HotpotQA |
|---|---|---|---|
| Avg Entities | 170.6 | 10.1 | 39.3 |
| Avg. Relations | 73.2 | 3.82 | 12.7 |

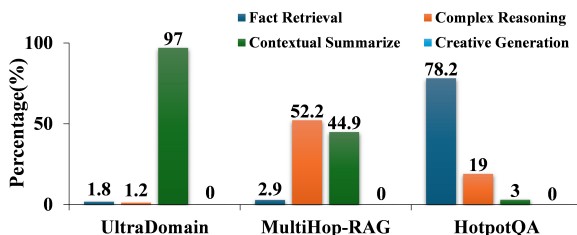

Figure 2: Distribution of question difficulty levels.

Third, current benchmarks fall short in evaluating GraphRAG since their evaluation metrics focus solely on the final outputs, answer accuracy or fluency, while treating GraphRAG's internal processes (graph construction, retrieval, and generation) as black boxes. Such evaluations can hardly measure

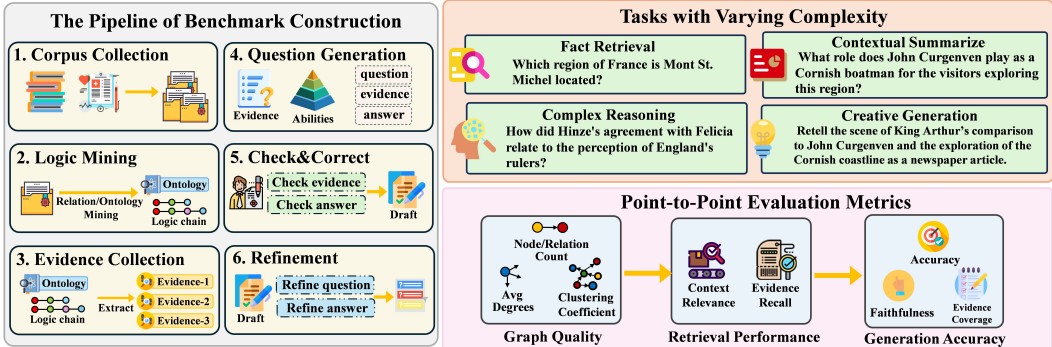

Figure 3: The overall framework of GraphRAG-Bench. It consists of three key components: (i) pipeline of benchmark construction (left), (ii) task classification by difficulty (upper right), and (iii) a multi-stage evaluation framework covering indexing, retrieval, and generation (lower right).

how graph structures contribute to the retrieval and reasoning processes. To truly assess GraphRAG, a more holistic evaluation is necessary, encompassing the entire pipeline. This includes examining the efficiency of graph construction and the quality of the resulting graph, the structure and relevance of the knowledge retrieved via the graph, and finally, the faithfulness of the generated answer to this graph-derived context. Such a comprehensive view is essential to understand the actual impact and benefits of graph structures within retrieval augmented generation systems. Detailed corpus statistics and analysis for these benchmarks can be found in Section E of the Appendix.

## 3  GRAPHRAG-BENCH

In this section, we present GraphRAG-Bench, a novel benchmark specifically designed to assess GraphRAG systems through comprehensive task hierarchies and structured knowledge integration. Specifically, GraphRAG-Bench consists a comprehensive dataset with (i) tasks of increasing difficulty, covering fact retrieval, multi-hop reasoning, Contextual Summarize, and creative generation, and (ii) real-world corpora with different information density, and (iii) a systematic evaluation across the entire pipeline, from graph construction and knowledge retrieval to final generation.

### 3.1  TASK FORMULATION

Traditional benchmarks focus on tasks with simple fact retrieval or linear multi-hop reasoning, where answers depend on linking concepts or facts across a limited set of documents. While these tasks test a model's ability to locate scattered information (retrieval difficulty), they do not reflect real-world scenarios demanding complex logical synthesis (reasoning complexity). Our benchmark addresses this gap by designing four different tasks that progressively scale both retrieval difficulty and reasoning complexity. As shown in Table 1, these four tasks ensure more comprehensive evaluation: lower-level tasks validate retrieval capability, while higher levels assess reasoning depth, ensuring models balance precise fact extraction with clear contextual comprehension.

### 3.2  DATASET CONSTRUCTION

**Corpus collection.** Existing datasets often derive from generic sources like Wikipedia or news articles, which, while broadly accessible, lack explicit logical connections and structured domain expertise, unable to evaluate systems that require reasoning over implicit relationships or contextual hierarchies. We address these issues by (i) integrating tightly structured domain data from NCCN medical guidelines to embed explicit hierarchies and standardized protocols, which provide dense conceptual relationships (e.g., treatment protocols linking symptoms, drugs, and outcomes) at scales exceeding typical domain corpora, and (ii) collecting loosely organized texts (pre-20th-century novels) from Gutenberg library to simulate real-world documents with implicit, non-linear narratives, ensuring the corpus reflects the complexity of unstructured knowledge while minimizing pretraining contamination. This combination ensures the corpus balances unstructured, real-world ambiguity with

domain-specific hierarchies, enabling rigorous evaluation of both retrieval robustness and reasoning depth. We include more details about these datasets (Novel and Medical Datasets) in Appendix C.

**Logic and evidence extraction.** To overcome the superficial treatment of reasoning in existing benchmarks, where multi-hop queries often reduce to linear fact retrieval, our framework systematically transforms raw text into structured domain ontologies. These ontologies preserve not only entities but also their contextual relationships and hierarchical dependencies, enabling the extraction of fine-grained evidence that reflects both localized factual clusters and interconnected reasoning chains. Where prior work struggles to represent latent logical synthesis (e.g., inferring causal pathways from dispersed market factors), our evidence extraction process isolates self-contained subgraphs for basic retrieval while reconstructing multi-hop relational sequences that expose deeper inferential patterns.

**Question generation.** we generate the questions according to the complexity of the underlying evidence. Rather than treating difficulty as a function of factual scarcity or hop count, we calibrate questions by progressively integrating evidence types, from isolated subgraph fragments for retrieval tasks to global topology-aware reasoning for synthetic reasoning. This ensures that complex questions necessitate not merely aggregating discrete facts but synthesizing contextual hierarchies and domain-specific ontologies. By anchoring questions in structured evidence packages that mirror real-world knowledge interdependencies, our benchmark evaluates how models derive insights from both explicit logical frameworks and unstructured contextual ambiguity, thereby addressing the critical gap in assessing reasoning depth beyond simple retrieval.

**Relevance check and refinement.** To ensure the accuracy and practical relevance of the dataset, we implemented rigorous validation and refinement processes after initial construction. Full methodological details are provided in Appendix C, while the visualization of our datasets is in Appendix E.

## 3.3 EVALUATION METRICS

Existing benchmarks primarily focus on the accuracy or fluency of final outputs, while how to measure the graph's contribution to the retrieval and reasoning processes remains an open challenge. To address this, we design stage-specific metrics that evaluate the entire workflow from graph construction and retrieval to final generation. In this section, we introduce these metrics accordingly.

**Graph Quality.** GraphRAG constructs graphs to represent the domain concepts and their relations, which enables structured and effective knowledge organization. To evaluate its effectiveness, we design structure-based metrics to assess the quality of graphs built in different GraphRAG.

• **NODE COUNT** quantifies the number of entities extracted during knowledge graph construction. Higher values imply broader domain coverage and finer-grained knowledge representation.

• **EDGE COUNT** measures the number of relations among entities. Higher values indicate denser semantic connectivity, facilitating multi-hop reasoning and complex query handling.

• **AVERAGE DEGREE** captures global connectivity by averaging the number of edges per node. Higher values of AVERAGE DEGREE indicate more integrated knowledge representations, enabling efficient cross-node traversal. It is computed as:

$$\text{AVERAGE DEGREE} = \frac{1}{|\mathcal{V}|} \sum_{v \in \mathcal{V}} \deg(v), \tag{1}$$

where $\mathcal{V}$ is the set of nodes, and $\deg(v)$ is the degree of node $v$.

• **AVERAGE CLUSTERING COEFFICIENT** evaluates local neighborhood connectivity via triad completion. Higher values, common in domain-specific clusters (e.g., disease–treatment–symptom in medical graphs), indicate coherent subgraphs that support localized reasoning. It can be obtained by:

$$\text{AVERAGE CLUSTERING COEFFICIENT} = \frac{1}{|\mathcal{V}|} \sum_{v \in \mathcal{V}} \text{C}(v), \quad \text{C}(v) = \frac{2 \cdot T(v)}{\deg(v) \cdot (\deg(v) - 1)}. \tag{2}$$

Here, $\text{C}(v)$ is the clustering coefficient of node $v$, with $T(v)$ denoting its centered triangle number.

**Retrieval Performance.** To evaluate the retrieval performance of GraphRAG, we argue that an effective system should not only ensure the completeness of retrieved information (i.e., high recall) but

Table 3: Results of Generate Evaluation using GPT-4o-mini, covering tasks of varying complexity.

| Category | Model | Fact Retrieval ACC | Fact Retrieval ROUGE-L | Complex Reasoning ACC | Complex Reasoning ROUGE-L | Contextual Summarize ACC | Contextual Summarize Cov | Creative Generation ACC | Creative Generation FS | Creative Generation Cov |
|---|---|---|---|---|---|---|---|---|---|---|
| | | | | | *Novel Dataset* | | | | | |
| Basic RAG | RAG (w/o rerank) | 58.76 | 37.35 | 41.35 | 15.12 | 50.08 | 82.53 | 41.52 | 47.46 | 37.84 |
| | RAG (w rerank) | 60.92 | 36.08 | 42.93 | 15.39 | 51.30 | 83.64 | 38.26 | 49.21 | 40.04 |
| Graph RAG | MS-GraphRAG (Edge et al., 2024) | 49.29 | 26.11 | 50.93 | 24.09 | 64.40 | 75.58 | 39.10 | 55.44 | 35.65 |
| | HippoRAG (Gutiérrez et al., 2024) | 52.93 | 26.65 | 38.52 | 11.16 | 48.70 | 85.55 | 38.85 | 71.53 | 38.97 |
| | HippoRAG2 (Gutiérrez et al., 2025) | 60.14 | 31.35 | 53.38 | 33.42 | 64.10 | 70.84 | 48.28 | 49.84 | 30.95 |
| | LightRAG (Guo et al., 2024) | 58.62 | 35.72 | 49.07 | 24.16 | 48.85 | 63.05 | 23.80 | 57.28 | 25.01 |
| | Fast-GraphRAG (CircleMind-AI, 2024) | 56.95 | 35.90 | 48.55 | 21.12 | 56.41 | 80.82 | 46.18 | 57.19 | 36.99 |
| | RAPTOR (Sarthi et al., 2024) | 49.25 | 23.74 | 38.59 | 11.66 | 47.10 | 82.33 | 38.01 | 70.85 | 35.88 |
| | Lazy-GraphRAG (Darren Edge, 2024) | 51.65 | 36.97 | 49.22 | 23.48 | 58.29 | 76.94 | 43.23 | 50.69 | 39.74 |
| | | | | | *Medical Dataset* | | | | | |
| Basic RAG | RAG (w/o rerank) | 63.72 | 29.21 | 57.61 | 13.98 | 63.72 | 77.34 | 58.94 | 35.88 | 57.87 |
| | RAG (w/ rerank) | 64.73 | 30.75 | 58.64 | 15.57 | 65.75 | 78.54 | 60.61 | 36.74 | 58.72 |
| GraphRAG | MS-GraphRAG (Edge et al., 2024) | 38.63 | 26.80 | 47.04 | 21.99 | 41.87 | 22.98 | 53.11 | 32.65 | 39.42 |
| | HippoRAG (Gutiérrez et al., 2024) | 56.14 | 20.95 | 55.87 | 13.57 | 59.86 | 62.73 | 64.43 | 69.21 | 65.56 |
| | HippoRAG2 (Gutiérrez et al., 2025) | 66.28 | 36.69 | 61.98 | 36.97 | 63.08 | 46.13 | 68.05 | 58.78 | 51.54 |
| | LightRAG (Guo et al., 2024) | 63.32 | 37.19 | 61.32 | 24.98 | 63.14 | 51.16 | 67.91 | 78.76 | 51.58 |
| | Fast-GraphRAG (CircleMind-AI, 2024) | 60.93 | 31.04 | 61.73 | 21.37 | 67.88 | 52.07 | 65.93 | 56.07 | 44.73 |
| | RAPTOR (Sarthi et al., 2024) | 54.07 | 17.93 | 53.20 | 11.73 | 58.73 | 78.28 | 62.38 | 59.98 | 63.63 |
| | Lazy-GraphRAG (Darren Edge, 2024) | 60.25 | 31.66 | 47.82 | 22.68 | 57.28 | 55.92 | 62.22 | 30.95 | 43.79 |

also reduce irrelevant content (i.e., high relevance). We introduce two corresponding retrieval-quality-based metrics: 1) CONTEXT RELEVANCE measures how well the retrieved content aligns with the question's intent by calculating the semantic similarity between the question and the retrieved context. 2) EVIDENCE RECALL measures retrieval completeness by assessing whether all critical components required to correctly answer the question are captured. Details are provided in Appendix F.

**Generation Accuracy.** After retrieval, a GraphRAG system is expected to generate accurate answers based on the retrieved contexts. To evaluate the quality of the generation, we introduce four key metrics: 1) LEXICAL OVERLAP: Measures word-level similarity between the generated and reference answers using longest common subsequence matching. 2) ANSWER ACCURACY: Assesses both semantic similarity and factual consistency with the reference answer. 3) FAITHFULNESS: Evaluates whether the relevant knowledge points in a long-form answer are faithful to the given context. 4) EVIDENCE COVERAGE: Measures whether the answer adequately covers all knowledge relevant to the question. We provide details of these widely used metrics in Appendix F.

## 4 EXPERIMENT

This section evaluates GraphRAG against RAG through comprehensive experiments on our new benchmarks. We aim to address the following research questions: **Q1** (Generation Accuracy): How does GraphRAG perform compared to RAG on our benchmark? **Q2** (Retrieval Performance): Does GraphRAG retrieve higher-quality and less redundant information in the retrieval process? **Q3** (Graph complexity): Does the constructed graph correctly organize the underlying knowledge? **Q4** (Efficiency): Does GraphRAG introduce significant token overhead during retrieval?

### 4.1 GENERATION ACCURACY (Q1)

To address Q1, we evaluate seven representative GraphRAG frameworks on our benchmark, using tailored metrics for different question types. For Type 1 (retrieval) and Type 2 (reasoning) questions, we assess answer quality with ROUGE scores and accuracy. For Type 3 (summarization) questions, we introduce evidence coverage to measure the comprehensiveness of the generated answers. For Type 4 (creative generation) questions, we use faithfulness to assess factual consistency. Main results in Table 3 and Appendix G lead to following observations:

**Obs.1 . Basic RAG Matches GraphRAG in simple fact retrieval task:** basic RAG is comparable to or outperforms GraphRAG in simple fact retrieval tasks that does not require com-

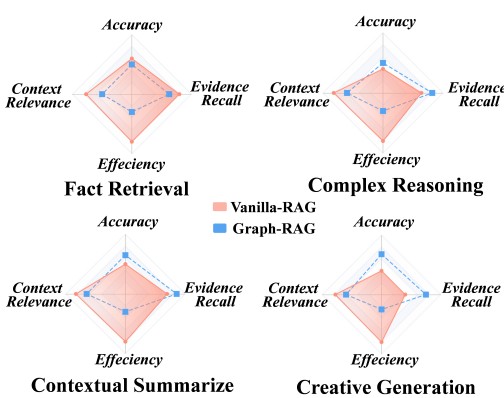

Figure 4: Retrieval and generation performance of RAG and GraphRAG across four different tasks.

Table 4: Results of Retrieval performance using GPT-4o-mini, covering tasks of varying complexity.

| Category | Model | Fact Retrieval | | Complex Reasoning | | Contextual Summarize | | Creative Generation | |
|---|---|---|---|---|---|---|---|---|---|
| | | Recall | Relevance | Recall | Relevance | Recall | Relevance | Recall | Relevance |
| *Novel Dataset* | | | | | | | | | |
| Basic RAG | RAG (w/o rerank) | 61.37 | 74.66 | 59.80 | 80.82 | 69.08 | 80.05 | 32.48 | 82.84 |
| | RAG (w/ rerank) | 83.21 | 77.77 | 64.47 | 82.08 | 73.38 | 83.10 | 39.59 | 78.73 |
| Graph RAG | MS-GraphRAG(Edge et al., 2024) | 61.04 | 27.30 | 73.03 | 39.09 | 82.02 | 43.13 | 53.55 | 35.07 |
| | HippoRAG(Gutiérrez et al., 2024) | 80.44 | 56.34 | 87.91 | 58.75 | 90.95 | 59.46 | 65.51 | 46.64 |
| | HippoRAG2(Gutiérrez et al., 2025) | 70.29 | 79.25 | 69.77 | 85.75 | 82.50 | 87.82 | 42.18 | 79.10 |
| | LightRAG(Guo et al., 2024) | 73.69 | 33.08 | 85.52 | 37.46 | 87.59 | 38.02 | 71.72 | 38.06 |
| | Fast-GraphRAG(CircleMind-AI, 2024) | 64.48 | 47.86 | 73.51 | 55.21 | 78.58 | 49.74 | 56.31 | 46.27 |
| | RAPTOR(Sarthi et al., 2024) | 62.14 | 54.08 | 67.80 | 61.26 | 75.79 | 63.00 | 58.66 | 58.46 |
| | Lazy-GraphRAG (Darren Edge, 2024) | 59.25 | 30.76 | 57.73 | 42.98 | 77.38 | 43.62 | 55.24 | 31.94 |
| *Medical Dataset* | | | | | | | | | |
| Basic RAG | RAG (w/o rerank) | 86.24 | 63.71 | 84.97 | 84.11 | 84.14 | 89.94 | 44.88 | 58.73 |
| | RAG (w rerank) | 87.83 | 64.73 | 86.49 | 85.56 | 85.87 | 91.35 | 45.23 | 60.50 |
| Graph RAG | MS-GraphRAG (Edge et al., 2024) | 38.06 | 05.67 | 61.32 | 04.25 | 59.66 | 05.24 | 66.59 | 02.76 |
| | HippoRAG (Gutiérrez et al., 2024) | 87.25 | 52.44 | 83.80 | 42.19 | 83.46 | 49.13 | 81.66 | 45.03 |
| | HippoRAG2 (Gutiérrez et al., 2025) | 78.70 | 87.96 | 77.00 | 80.94 | 77.40 | 86.85 | 61.12 | 78.64 |
| | LightRAG (Guo et al., 2024) | 80.32 | 41.27 | 82.91 | 42.79 | 85.71 | 43.11 | 81.34 | 45.17 |
| | Fast-GraphRAG (CircleMind-AI, 2024) | 66.82 | 45.86 | 74.93 | 38.80 | 77.27 | 47.58 | 62.99 | 25.15 |
| | RAPTOR (Sarthi et al., 2024) | 85.40 | 69.38 | 89.70 | 53.20 | 88.86 | 58.73 | 72.70 | 52.71 |
| | Lazy-GraphRAG (Darren Edge, 2024) | 74.29 | 19.90 | 78.65 | 17.50 | 78.72 | 21.35 | 83.41 | 15.09 |

plex reasoning across connected concepts. This
suggests that in less complex scenarios, basic
RAG's straightforward retrieval method is sufficient, while GraphRAG's extra graph-based processing
may introduce redundant or noisy information for simpler queries, degrading answer quality.

**Obs.2 . GraphRAG excels in complex tasks**: GraphRAG models show a clear advantage in complex
reasoning, Contextual Summarize, and creative generation. This is intuitive, as these tasks require
bridging the complex relations among multiple concepts, which is naturally a graph structure.

**Obs.3 . GraphRAG ensures greater factual reliability in creative tasks**: RAPTOR scores highest
in faithfulness (70.9%) on the novel dataset, though RAG covers more evidence (40.0%), likely
because GraphRAG's fragmented knowledge retrieval and complicates broad scope generation. This
trade-off highlights GraphRAG's strength in precision but limitations in wide-ranging synthesis.

## 4.2 RETRIEVAL PERFORMANCE (Q2)

To quantitatively compare the retrieval effectiveness of the two paradigms, we adopt two comple-
mentary metrics: Evidence Recall, which measures how completely the retrieved context covers the
gold evidence, and Context Relevance, which measures the semantic alignment between the retrieved
content and the input query. As shown in Table 4 and Appendix G, we have the following observtions:

**Obs.4 . RAG excels at retrieving discrete facts for simple questions that do not require complex
logics**, achieving 83.2% Evidence Recall on the novel dataset (vs. HippoRAG2's best Context
Relevance). Medical dataset results confirm this pattern, suggesting relevant evidence for Level 1
questions typically resides in single passages. It is because the graph used in GraphRAG introduces
several logically relevant but redundant information in these scenarios.

**Obs.5 . GraphRAG's advantages emerge clearly as questions grow more complex.** For Level
2-3 questions on the novel dataset, HippoRAG achieves remarkable Evidence Recall (87.9-90.9%),
while HippoRAG2 leads in Context Relevance (85.8-87.8%). Medical dataset results reinforce this
trend, demonstrating GraphRAG's unique ability to connect information across distant text segments,
crucial for multi-hop reasoning and comprehensive summarization.

**Obs.6 . RAG and GraphRAG show a trade-off on creative tasks requiring broad knowledge
synthesis.** Global-GraphRAG achieves superior Evidence Recall (83.1%), though RAG maintains
better Context Relevance (78.8%). While GraphRAG accesses more relevant information overall, its
retrieval approach naturally introduces some redundancy compared to RAG's more focused results.

## 4.3 GRAPH COMPLEXITY (Q3)

During the indexing phase, GraphRAG extracts entities and relations from the corpus to construct a
knowledge graph. By indexing over the graph structure, GraphRAG establishes logical and semantic

connections between the knowledge graph and the original context, resulting in a well-structured and knowledge-complete index graph. To reveal the structural characteristics of the index graph and highlight differences introduced by various GraphRAG, we introduce the following metrics: number of nodes, number of edges, average degree, and average clustering coefficient.

**Obs.7 . The index graphs generated by different GraphRAG implementations demonstrate substantial structural variation.**

As illustrated in Figure 5, HippoRAG2 produces significantly denser graphs, with node and edge counts that far surpass other frameworks. Specifically, on the novel dataset, HippoRAG2 has an average of 2,310 edges and 523 nodes, while on the medical dataset, it has an average of 3,979 edges and 598 nodes (per 10k corpus tokens). This enhanced graph density improves both information connectivity and coverage, ultimately contributing to superior retrieval and generation capabilities. This observation is consistent with the retrieval performance, which shows that HippoRAG2 achieves higher recall than other baselines.

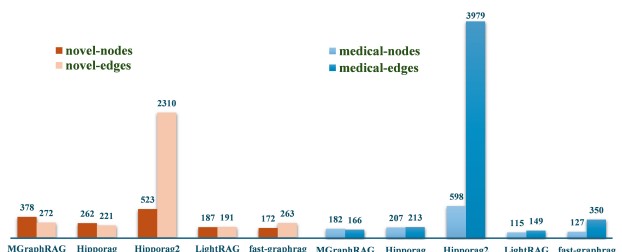

Figure 5: The relational structure of different methods.

Table 5: The Graph statistics across RAG methods.

| Metric | MS-GraphRAG | HippoRAG2 | LightRAG | Fast-GraphRAG | HippoRAG |
|---|---|---|---|---|---|
| **Novel Dataset** | | | | | |
| Average Degree | 1.48 | 8.75 | 2.10 | 3.19 | 1.73 |
| Avg. Clust. Coeff | 0.315 | 0.657 | 0.212 | 0.324 | 0.100 |
| **Medical Dataset** | | | | | |
| Average Degree | 1.82 | 13.31 | 2.58 | 5.50 | 2.06 |
| Avg. Clust. Coeff | 0.300 | 0.497 | 0.139 | 0.347 | 0.087 |

## 4.4 EFFICIENCY (Q4)

GraphRAG retrieves relevant knowledge by traversing the constructed graph. While this approach allows for more structured knowledge organization, it can also lead to a substantial increase in token cost. To better understand the associated efficiency and cost implications, we conduct a dedicated analysis on prompt statistics across different GraphRAG models.

**Obs.8 . Compared to vanilla RAG, GraphRAG significantly increases prompt length** due to the additional steps involved in knowledge retrieval and graph-based aggregation. Specifically, MS-GraphRAG(global), which incorporates a community-summarization mechanism, reaches a prompt size of up to $4 \times 10^4$ tokens. LightRAG also produces lengthy prompts ($\approx 10^4$ tokens). In contrast, HippoRAG2 maintains a more compact prompt size ($\approx 10^3$ tokens), showing better efficiency. These results highlight that GraphRAG's structured pipeline incurs non-trivial token overhead.

Table 6: Ave token cost of different GraphRAG (Part 1).

| Avg Tokens | V-RAG | MS-GraphRAG(local) | MS-GraphRAG(global) | HippoRAG2 |
|---|---|---|---|---|
| Novel | 879 | 38707 | 331375 | 1008 |
| Medical | 954 | 39821 | 332881 | 1020 |

Table 7: Ave token cost of different GraphRAG (Part 2).

| Avg Tokens | LightRAG | Fast-GraphRAG | RAPTOR | HippoRAG |
|---|---|---|---|---|
| Novel | 100832 | 4204 | 3441 | 7208 |
| Medical | 100310 | 4298 | 3510 | 7342 |

**Obs.9 . As task complexity and the number of required knowledge points increase, GraphRAG's prompt length exhibits a clear upward trend**. Notably, MS-GraphRAG(global)'s prompt size expands from 7,800 to 40,000 tokens across tasks of increasing difficulty. This excessive token accumulation often introduces redundant information, which in turn degrades context relevance during retrieval. These findings underscore a critical trade-off: while GraphRAG improves retrieval breadth, it may also introduce noisy context due to prompt inflation, especially in complex tasks.

## 5 CONCLUSION

Graph-based Retrieval-Augmented Generation (GraphRAG) emerges as a pioneering approach that introduces graph structures to explicitly model entity relationships and hierarchical dependencies,

enabling more coherent and effective knowledge retrieval. Despite its conceptual promise, empirical studies report that GraphRAG often fails to outperform vanilla RAG on many NLP tasks, raising questions about its real-world effectiveness. This paper systematically investigates when and why GraphRAG succeeds, offering practical guidelines for its application. Specifically, we first conduct an extensive analysis on existing benchmark datasets and identify that they inadequately assess GraphRAG due to the lack of domain-specific corpora and oversimplified task granularity. Based on the findings, we propose a comprehensive benchmark designed to evaluate GraphRAG models in terms of hierarchical knowledge retrieval and deep contextual reasoning.

## ETHICS STATEMENT

The benchmark introduced in this paper is constructed from publicly available internet data. We have taken care to ensure our data collection process respects user privacy by filtering for personally identifiable information and complies with the terms of the source platforms. All models used in our evaluation are open-source, promoting transparency and reproducibility. While we have focused on the technical aspects of reasoning, we advise users to be mindful of potential societal biases that may exist in the source data and to consider the broader context of their applications.

## REPRODUCIBILITY STATEMENT

To ensure the reproducibility of our research, we have made our dataset, evaluation code, and detailed experimental settings publicly available. We have open-sourced the original data for our benchmark and all the code required to replicate the results presented in this paper. These resources are available in the repository at `https://github.com/GraphRAG-Bench/GraphRAG-Benchmark`. Furthermore, we provide a detailed description of our dataset construction process in Appendix C. The specific hyperparameter settings used for all baseline models evaluated in our experiments are also fully documented in Appendix H.2.

## ACKNOWLEDGMENTS

This work was supported by the Natural Science Foundation of Fujian Province of China (No. 2024J011001) and the Public Technology Service Platform Project of Xiamen (No. 3502Z20231043). We thank the anonymous reviewers for their insightful comments.

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

# Appendix

## CONTENTS

# A    FREQUENTLY ASKED QUESTIONS (FAQS)

## A.1    CODE AND LEADERBOARD

To promote transparency and reproducibility, we have uploaded our code to GitHub at `https://github.com/GraphRAG-Bench/GraphRAG-Benchmark`. This repository includes the source code and scripts for evaluation, ensuring that researchers have full access to the resources required to reproduce and extend our work. In addition, we will continue to maintain and update the repository to reflect future improvements. Besides that, the leaderboard visualization is provided in Figure 6. We have also updated the related resources to the leaderboard.

🏆 **Leaderboard**

**GraphRAG-Bench (Novel)**

| Model↕ | Rank↕ | Average↕ | Date↕ | Fact Retrieval | | Complex Reasoning | | Contextual Summarize | | Creative Generation | | | Link |
|---|---|---|---|---|---|---|---|---|---|---|---|---|---|
| | | | | ACC↕ | ROUGE-L↕ | ACC↕ | ROUGE-L↕ | ACC↕ | Cov↕ | ACC↕ | FS↕ | Cov↕ | |
| HippoRAG2 | 1 | 56.48 | 2025-02 | 60.14 | 31.35 | 53.38 | 33.42 | 64.1 | 70.84 | 48.28 | 49.84 | 30.95 | 🔗 |
| Fast-GraphRAG | 2 | 52.02 | 2024-10 | 56.95 | 35.9 | 48.55 | 21.12 | 56.41 | 80.82 | 46.18 | 57.19 | 36.99 | 🔗 |
| MS-GraphRAG (local) | 3 | 50.93 | 2024-04 | 49.29 | 26.11 | 50.93 | 24.09 | 64.4 | 75.58 | 39.1 | 55.44 | 35.65 | 🔗 |
| Lazy-GraphRAG | 4 | 50.59 | 2024-09 | 51.65 | 36.97 | 49.22 | 23.48 | 58.29 | 76.94 | 43.23 | 50.69 | 39.74 | 🔗 |
| StructRAG | 5 | 49.13 | 2024-10 | 53.84 | 26.73 | 46.27 | 23.49 | 54.28 | 63.56 | 42.16 | 52.68 | 36.75 | 🔗 |
| RAG (w rerank) | 6 | 48.35 | - | 60.92 | 36.08 | 42.93 | 15.39 | 51.3 | 83.64 | 38.26 | 49.21 | 40.04 | 🔗 |
| KGP | 7 | 48.01 | 2023-08 | 54.15 | 24.73 | 46.31 | 16.91 | 51.21 | 64.34 | 40.37 | 52.55 | 34.65 | 🔗 |
| RAG (w/o rerank) | 8 | 47.93 | - | 58.76 | 37.35 | 41.35 | 15.12 | 50.08 | 82.53 | 41.52 | 47.46 | 37.84 | 🔗 |
| KET-RAG | 9 | 47.62 | 2025-02 | 55.39 | 27.39 | 36.59 | 25.98 | 52.47 | 69.24 | 46.03 | 36.72 | 33.68 | 🔗 |
| LightRAG | 10 | 45.09 | 2024-10 | 58.62 | 35.72 | 49.07 | 24.16 | 48.85 | 63.05 | 23.8 | 57.28 | 25.01 | 🔗 |
| HippoRAG | 11 | 44.75 | 2024-05 | 52.93 | 26.65 | 38.52 | 11.16 | 48.7 | 85.55 | 38.85 | 71.53 | 38.97 | 🔗 |
| MS-GraphRAG (global) | 12 | 44.52 | 2024-04 | 36.92 | 17.32 | 43.17 | 15.12 | 56.87 | 80.55 | 41.11 | 75.15 | 30.34 | 🔗 |
| RAPTOR | 13 | 43.24 | 2024-01 | 49.25 | 23.74 | 38.59 | 11.66 | 47.1 | 82.33 | 38.01 | 70.85 | 35.88 | 🔗 |

**GraphRAG-Bench (Medical)**

| Model↕ | Rank↕ | Average↕ | Date↕ | Fact Retrieval | | Complex Reasoning | | Contextual Summarize | | Creative Generation | | | Link |
|---|---|---|---|---|---|---|---|---|---|---|---|---|---|
| | | | | ACC↕ | ROUGE-L↕ | ACC↕ | ROUGE-L↕ | ACC↕ | Cov↕ | ACC↕ | FS↕ | Cov↕ | |
| HippoRAG2 | 1 | 64.85 | 2025-02 | 66.28 | 36.69 | 61.98 | 36.97 | 63.08 | 46.13 | 68.05 | 58.78 | 51.54 | 🔗 |
| Fast-GraphRAG | 2 | 64.12 | 2024-10 | 60.93 | 31.04 | 61.73 | 21.37 | 67.88 | 52.07 | 65.93 | 56.07 | 44.73 | 🔗 |
| LightRAG | 3 | 62.59 | 2024-10 | 63.32 | 37.19 | 61.32 | 24.98 | 63.14 | 51.16 | 67.91 | 78.76 | 51.58 | 🔗 |
| RAG (w/ rerank) | 4 | 62.43 | - | 64.73 | 30.75 | 58.64 | 15.57 | 65.75 | 78.54 | 60.61 | 36.74 | 58.72 | 🔗 |
| RAG (w/o rerank) | 5 | 61.00 | - | 63.72 | 29.21 | 57.61 | 13.98 | 63.72 | 77.34 | 58.94 | 35.88 | 57.87 | 🔗 |
| HippoRAG | 6 | 59.08 | 2024-05 | 56.14 | 20.95 | 55.87 | 13.57 | 59.86 | 62.73 | 64.43 | 69.21 | 65.56 | 🔗 |
| StructRAG | 7 | 58.56 | 2024-10 | 55.38 | 27.53 | 56.17 | 22.79 | 62.48 | 65.66 | 60.21 | 42.35 | 45.76 | 🔗 |
| RAPTOR | 8 | 57.10 | 2024-01 | 54.07 | 17.93 | 53.2 | 11.73 | 58.73 | 78.28 | 62.38 | 59.98 | 63.63 | 🔗 |
| Lazy-GraphRAG | 9 | 56.89 | 2024-09 | 60.25 | 31.66 | 47.82 | 22.68 | 57.28 | 55.92 | 62.22 | 30.95 | 43.79 | 🔗 |
| KGP | 10 | 56.33 | 2023-08 | 55.53 | 21.34 | 51.53 | 11.69 | 54.51 | 62.4 | 63.77 | 45.25 | 35.55 | 🔗 |
| KET-RAG | 11 | 47.05 | 2025-02 | 60.35 | 31.99 | 39.56 | 19.52 | 45.27 | 29.04 | 43.04 | 33.67 | 31.93 | 🔗 |
| MS-GraphRAG (local) | 12 | 45.16 | 2024-04 | 38.63 | 26.8 | 47.04 | 21.99 | 41.87 | 22.98 | 53.11 | 32.65 | 39.42 | 🔗 |
| MS-GraphRAG (global) | 13 | 28.56 | 2024-04 | 16.42 | 46 | 15.61 | 52.75 | 19.82 | - | 20.81 | - | 13.64 | 🔗 |

Figure 6: Overview of the leaderboard, ranked by average generation performance (ACC).

## A.2    WHY IS IT IMPORTANT TO INCLUDE TASKS WITH VARYING COMPLEXITY?

Task complexity is critical for assessing GraphRAG models because their core value lies in navigating interconnected knowledge structures and synthesizing latent logical relationships. Real-world challenges demand more than locating scattered facts since they require integrating hierarchical domain expertise with ambiguous, context-dependent narratives to form coherent insights. By evaluating models on tasks of varying complexity, from factual retrieval to creative generation, we expose whether they truly leverage graph structures to reason like humans: inferring causality, resolving conflicting contexts, and extrapolating insights beyond explicit data. Without measuring how models

handle complexity, evaluations risk overestimating their utility for real applications, where success hinges on synthesizing interconnected concepts, not just retrieving them. Complexity-aware assessment ensures GraphRAG systems are validated on their ability to map, traverse, and reason through domain-specific ontologies, the very features that distinguish them from traditional RAG frameworks.

### A.3    HOW TO CONTROL THE COMPLEXITY OF EACH TASK?

GraphRAG-Bench leverages structural information to control complexity. We begin by extracting information from corpora to build ontologies or knowledge graphs. Taking the Novel dataset as an example, we first construct a knowledge graph from the corpus, then form logic chains using relevant triples from this graph. Then, we control difficulty by adjusting the number of involved triples and the information span they cover. Specifically, we define question complexity through two dimensions: Knowledge Breadth (measured by the count of triples required to answer a question) and Reasoning Depth (measured by the number of inference hops between these triples). The relevant statistical results are shown in Table 8.

Table 8: Problem Complexity Statistics of GraphRAG-Bench.

| Problem Complexity | Fact Retrieval | Complex Reasoning | Contextual Summarize | Creative Generation |
|---|---|---|---|---|
| *Novel Dataset* | | | | |
| Knowledge Breadth | 1.40 | 2.60 | 3.51 | 7.11 |
| Reasoning Depth | 1.69 | 6.25 | 4.64 | 7.81 |
| *Medical Dataset* | | | | |
| Knowledge Breadth | 1.25 | 3.45 | 5.1 | 10.14 |
| Reasoning Depth | 1.82 | 5.23 | 4.27 | 8.27 |

### A.4    WHY CONSTRUCT TWO DATASETS FROM BOTH NOVELS AND THE MEDICAL CORPUS?

Including two distinct datasets, medical guidelines and unstructured novels, is essential to evaluate GraphRAG models under conditions that mirror real-world knowledge ecosystems. Medical corpora provide explicit, hierarchical relationships, testing a model's ability to navigate rigid domain logic and standardized protocols. Conversely, Novel corpora introduce implicit, context-dependent dependencies, like socio-historical factors shaping character decisions, challenging models to infer latent connections without predefined rules. This approach ensures the benchmark assesses both precision in following formal hierarchies and adaptability in interpreting ambiguous, open-ended contexts, critical for applications where models must integrate structured expertise with unstructured, real-world narratives.

### A.5    WHY NOT USE OTHER FORMATS FOR QUESTION CONSTRUCTION?

In this paper, we aim to figure out in which scenarios do graph structures provide measurable benefits for RAG systems. In other words, we care more about the retrieval and reasoning complexity of task, instead of the task type or question format. It is because the specific task type or format (like QA, multiple-choice, or fact-checking) doesn't really change: 1) reasoning difficulty: the reasoning steps the model needs to take, 2) retrieval difficulty: how to locate scattered information from the corpus.

Given a question of "What famous universities are in San Diego?". Whether this is asked as: An open QA question, or A multiple-choice question (e.g., "Is UCSD in San Diego? A) Yes B) No"), or A fact-checking task (e.g., "Check this: UCSD has a campus in San Diego")... doesn't change the core need: The RAG system must still find information about San Diego and reason to answer correctly.

### A.6    WHAT IS THE ADVANTAGE OF USING STAGE-SPECIFIC EVALUATION METRICS?

Stage-specific evaluation metrics are crucial because they provide granular insights into how well a GraphRAG model performs at each phase of its pipeline, rather than relying solely on end-to-end output metrics like answer accuracy. Traditional benchmarks often treat the entire process as a "black box", obscuring whether failures stem from flawed knowledge graph construction, suboptimal

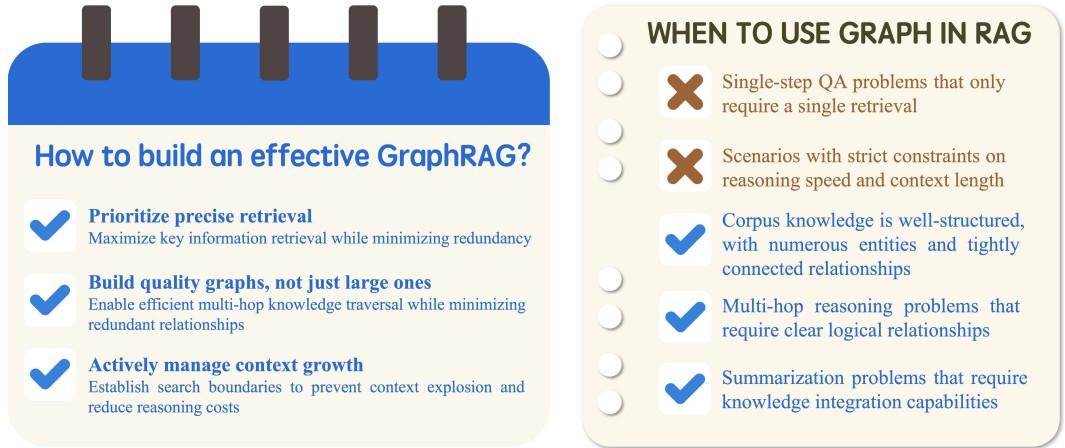

Figure 7: An overview of building an effective GraphRAG system. It consists of two key parts: (i) the crucial principles for system design (left), and (ii) the most suitable application scenarios (right).

retrieval, or weak reasoning. By designing metrics tailored to individual stages, such as graph completeness during logic mining, retrieval relevance in evidence collection, or contextual coherence in question generation, we isolate and diagnose weaknesses in specific components.

### A.7 COMPARISON WITH EXISTING BENCHMARKS AND ANALYSIS PAPERS.

Existing studies (Han et al., 2025; Zhou et al., 2025) focus on architectural comparisons using homogeneous datasets, missing how models synthesize hierarchical expertise and unstructured narratives. To this end, we propose GraphRAG-Bench, a comprehensive benchmark designed to evaluate GraphRAG models on deep reasoning. It features hybrid corpora (medical guidelines + novels) with tasks of increasing complexity and stage-specific metrics to expose why models fail, whether in graph construction, knowledge retrieval, or contextual synthesis. Leveraging this novel benchmark, we systematically investigate the conditions when GraphRAG surpasses traditional RAG systems and the underlying reasons for its success, offering guidelines for its practical application.

## B TAKEAWAY FINDINGS

In this paper, we not only build GraphRAG-Bench to evaluate existing GraphRAG systems, but more importantly, we provide insightful recommendations for future GraphRAG research, as illustrated in Figure 7.

PRIORITIZE PRECISE RETRIEVAL: Effective frameworks should focus on how to maximize key information retrieval while minimizing redundancy. It's critical to pinpoint the key facts needed to answer a question, while at the same time avoiding pulling in unnecessary details. This keeps the context clean and focused, which helps the model reason better and improves overall efficiency.

BUILD QUALITY GRAPHS, NOT JUST LARGE ONES: While GraphRAG forms knowledge graphs (entities/relationships) for efficient searching, more relationships $\neq$ better performance. Optimal graphs require tightly connected communities, which create denser structures rich in implicit multi-hop knowledge – enabling faster graph traversal.

ACTIVELY MANAGE CONTEXT GROWTH: Unlike traditional RAG (fixed context via vector search), GraphRAG retrieves entities, relationships, and raw text snippets, risking sudden context explosion and hig reasoning costs. Future solutions need search boundaries to curb context growth and significantly lower costs.

Table 9: Results of Generation Evaluation using GPT-4o-mini, covering tasks of varying complexity.

| Category | Model | Fact Retrieval | | Complex Reasoning | | Contextual Summarize | | Creative Generation | | |
|---|---|---|---|---|---|---|---|---|---|---|
| | | ACC | ROUGE-L | ACC | ROUGE-L | ACC | Cov | ACC | FS | Cov |
| | *Novel Dataset* | | | | | | | | | |
| Basic RAG | RAG (w/o rerank) | 58.76 | 37.35 | 41.35 | 15.12 | 50.08 | 82.53 | 41.52 | 47.46 | 37.84 |
| | RAG (w rerank) | 60.92 | 36.08 | 42.93 | 15.39 | 51.30 | 83.64 | 38.26 | 49.21 | 40.04 |
| Graph RAG | MS-GraphRAG (local) (Edge et al., 2024) | 49.29 | 26.11 | 50.93 | 24.09 | 64.40 | 75.58 | 39.10 | 55.44 | 35.65 |
| | MS-GraphRAG (global) (Edge et al., 2024) | 36.92 | 17.32 | 43.17 | 15.12 | 56.87 | 80.55 | 41.11 | 75.15 | 30.34 |
| | HippoRAG (Gutiérrez et al., 2024) | 52.93 | 26.65 | 38.52 | 11.16 | 48.70 | 85.55 | 38.85 | 71.53 | 38.97 |
| | HippoRAG2 (Gutiérrez et al., 2025) | 60.14 | 31.35 | 53.38 | 33.42 | 64.10 | 70.84 | 48.28 | 49.84 | 30.95 |
| | LightRAG (Guo et al., 2024) | 58.62 | 35.72 | 49.07 | 24.16 | 48.85 | 63.05 | 23.80 | 57.28 | 25.01 |
| | Fast-GraphRAG (CircleMind-AI, 2024) | 56.95 | 35.90 | 48.55 | 21.12 | 56.41 | 80.82 | 46.18 | 57.19 | 36.99 |
| | RAPTOR (Sarthi et al., 2024) | 49.25 | 23.74 | 38.59 | 11.66 | 47.10 | 82.33 | 38.01 | 70.85 | 35.88 |
| | Lazy-GraphRAG (Darren Edge, 2024) | 51.65 | 36.97 | 49.22 | 23.48 | 58.29 | 76.94 | 43.23 | 50.69 | 39.74 |
| | KGP (Wang et al., 2024b) | 54.15 | 24.73 | 46.31 | 16.91 | 51.21 | 64.34 | 40.37 | 52.55 | 34.65 |
| | StructRAG (Li et al., 2024) | 53.84 | 26.73 | 46.27 | 23.49 | 54.28 | 63.56 | 42.16 | 52.68 | 36.75 |
| | KET-RAG (Huang et al., 2025) | 55.39 | 27.39 | 36.59 | 25.98 | 52.47 | 69.24 | 46.03 | 36.72 | 33.68 |
| | *Medical Dataset* | | | | | | | | | |
| Basic RAG | RAG (w/o rerank) | 63.72 | 29.21 | 57.61 | 13.98 | 63.72 | 77.34 | 58.94 | 35.88 | 57.87 |
| | RAG (w/ rerank) | 64.73 | 30.75 | 58.64 | 15.57 | 65.75 | 78.54 | 60.61 | 36.74 | 58.72 |
| GraphRAG | MS-GraphRAG (local) (Edge et al., 2024) | 38.63 | 26.80 | 47.04 | 21.99 | 41.87 | 22.98 | 53.11 | 32.65 | 39.42 |
| | MS-GraphRAG (global) (Edge et al., 2024) | 16.42 | 46.00 | 15.61 | 52.75 | 19.82 | - | 20.81 | - | 13.64 |
| | HippoRAG (Gutiérrez et al., 2024) | 56.14 | 20.95 | 55.87 | 13.57 | 59.86 | 62.73 | 64.43 | 69.21 | 65.56 |
| | HippoRAG2 (Gutiérrez et al., 2025) | 66.28 | 36.69 | 61.98 | 36.97 | 63.08 | 46.13 | 68.05 | 58.78 | 51.54 |
| | LightRAG (Guo et al., 2024) | 63.32 | 37.19 | 61.32 | 24.98 | 63.14 | 51.16 | 67.91 | 78.76 | 51.58 |
| | Fast-GraphRAG (CircleMind-AI, 2024) | 60.93 | 31.04 | 61.73 | 21.37 | 67.88 | 52.07 | 65.93 | 56.07 | 44.73 |
| | RAPTOR (Sarthi et al., 2024) | 54.07 | 17.93 | 53.20 | 11.73 | 58.73 | 78.28 | 62.38 | 59.98 | 63.63 |
| | Lazy-GraphRAG (Darren Edge, 2024) | 60.25 | 31.66 | 47.82 | 22.68 | 57.28 | 55.92 | 62.22 | 30.95 | 43.79 |
| | KGP (Wang et al., 2024b) | 52.34 | 21.34 | 51.53 | 11.69 | 54.51 | 62.40 | 63.77 | 45.25 | 35.55 |
| | StructRAG (Li et al., 2024) | 55.38 | 27.53 | 56.17 | 22.79 | 62.48 | 65.66 | 60.21 | 42.35 | 45.76 |
| | KET-RAG (Huang et al., 2025) | 60.35 | 31.99 | 39.56 | 19.52 | 45.27 | 29.04 | 43.04 | 33.67 | 31.93 |

## C  BENCHMARK CONSTRUCTION

To address the critical limitations of existing RAG evaluation frameworks, we present a novel benchmark to assess GraphRAG systems through comprehensive task hierarchies and structured knowledge integration. Specifically, our benchmark is constructed through six stages that systematically integrate domain-specific logical hierarchies and contextual dependencies to enable precise control over question difficulty.

### C.1  CORPUS COLLECTION

Existing benchmarks often rely on corpora with inconsistent quality and inadequate information density, particularly in their inability to represent both loosely organized real-world knowledge and tightly structured domain-specific hierarchies. In contrast, our benchmark addresses this by constructing a corpus that integrates two complementary datasets: (i) **Medical Dataset**. We integrate domain data from the National Comprehensive Cancer Network (NCCN) clinical guidelines, which provide standardized treatment protocols, drug interaction hierarchies, and diagnostic criteria. (ii) **Novel Dataset**. We curated a collection of pre-20th-century novels (narrative fictions) from the Project Gutenberg library, prioritizing lesser-known works to minimize overlap with pretraining data of LLMs. These texts were selected based on their length and narrative ambiguity, ensuring they simulate real-world documents with non-linear, inferential dependencies. These two complementary datasets ensure the corpus balances unstructured, real-world ambiguity with domain-specific hierarchies, enabling rigorous evaluation of both retrieval robustness and reasoning depth.

### C.2  LOGIC MINING

Raw text alone lacks explicit representations of the latent relationships that define real-world reasoning, such as causality, hierarchy, or contradiction. Existing benchmarks often treat these relationships as implicit, leading to superficial evaluations of "multi-hop" queries as mere fact aggregation. To address this, we transform text into formalized ontologies using GPT-4.1, which codifies vertical hierarchies (e.g., symptom → diagnosis) and horizontal dependencies (e.g., socio-economic factors influencing medical outcomes). This ontology acts as a ground-truth map of domain logic, enabling precise identification of what constitutes a reasoning step and how concepts interrelate. By making these relationships explicit, we establish a measurable foundation for distinguishing factual retrieval from genuine logical synthesis, a prerequisite for controlling question difficulty.

Table 10: Results of Retrieval performance using GPT-4o-mini, covering tasks of varying complexity.

| Category | Model | Fact Retrieval Recall | Fact Retrieval Relevance | Complex Reasoning Recall | Complex Reasoning Relevance | Contextual Summarize Recall | Contextual Summarize Relevance | Creative Generation Recall | Creative Generation Relevance |
|---|---|---|---|---|---|---|---|---|---|
| | | *Novel Dataset* | | | | | | | |
| Basic RAG | RAG (w/o rerank) | 61.37 | 74.66 | 59.80 | 80.82 | 69.08 | 80.05 | 32.48 | 82.84 |
| | RAG (w/ rerank) | 83.21 | 77.77 | 64.47 | 82.08 | 73.38 | 83.10 | 39.59 | 78.73 |
| Graph RAG | MS-GraphRAG (local)(Edge et al., 2024) | 61.04 | 27.30 | 73.03 | 39.09 | 82.02 | 43.13 | 53.55 | 35.07 |
| | MS-GraphRAG (global)(Edge et al., 2024) | 42.27 | 09.37 | 86.68 | 14.36 | 89.69 | 15.35 | 83.14 | 19.40 |
| | HippoRAG(Gutiérrez et al., 2024) | 80.44 | 56.34 | 87.91 | 58.75 | 90.95 | 59.46 | 65.51 | 46.64 |
| | HippoRAG2(Gutiérrez et al., 2025) | 70.29 | 79.25 | 69.77 | 85.75 | 82.50 | 87.82 | 42.18 | 79.10 |
| | LightRAG(Guo et al., 2024) | 73.69 | 33.08 | 85.52 | 37.46 | 87.59 | 38.02 | 71.72 | 38.06 |
| | Fast-GraphRAG(CircleMind-AI, 2024) | 64.48 | 47.86 | 73.51 | 55.21 | 78.58 | 49.74 | 56.31 | 46.27 |
| | RAPTOR(Sarthi et al., 2024) | 62.14 | 54.08 | 67.80 | 61.26 | 75.79 | 63.00 | 58.66 | 58.46 |
| | Lazy-GraphRAG (Darren Edge, 2024) | 59.25 | 30.76 | 57.73 | 42.98 | 77.38 | 43.62 | 55.24 | 31.94 |
| | KGP (Wang et al., 2024b) | 55.71 | 23.71 | 63.51 | 31.96 | 61.54 | 64.20 | 67.57 | 35.52 |
| | StructRAG (Li et al., 2024) | 55.38 | 27.53 | 56.17 | 34.79 | 62.48 | 65.66 | 60.21 | 42.35 |
| | KET-RAG (Huang et al., 2025) | 63.55 | 39.11 | 56.93 | 32.59 | 67.35 | 39.05 | 53.40 | 36.74 |
| | | *Medical Dataset* | | | | | | | |
| Basic RAG | RAG (w/o rerank) | 86.24 | 63.71 | 84.97 | 84.11 | 84.14 | 89.94 | 44.88 | 58.73 |
| | RAG (w rerank) | 87.83 | 64.73 | 86.49 | 85.56 | 85.87 | 91.35 | 45.23 | 60.50 |
| Graph RAG | MS-GraphRAG (local)(Edge et al., 2024) | 38.06 | 05.67 | 61.32 | 04.25 | 59.66 | 05.24 | 66.59 | 02.76 |
| | MS-GraphRAG (global)(Edge et al., 2024) | 65.98 | 07.46 | 78.46 | 11.72 | 89.06 | 11.72 | 85.28 | 02.73 |
| | HippoRAG (Gutiérrez et al., 2024) | 87.25 | 52.44 | 83.80 | 42.19 | 83.46 | 49.13 | 81.66 | 45.03 |
| | HippoRAG2 (Gutiérrez et al., 2025) | 78.70 | 87.96 | 77.00 | 80.94 | 77.40 | 86.85 | 61.12 | 78.64 |
| | LightRAG (Guo et al., 2024) | 80.32 | 41.27 | 82.91 | 42.79 | 85.71 | 43.11 | 81.34 | 45.17 |
| | Fast-GraphRAG (CircleMind-AI, 2024) | 66.82 | 45.86 | 74.93 | 38.80 | 77.27 | 47.58 | 62.99 | 25.15 |
| | RAPTOR (Sarthi et al., 2024) | 85.40 | 69.38 | 89.70 | 53.20 | 88.86 | 58.73 | 72.70 | 52.71 |
| | Lazy-GraphRAG (Darren Edge, 2024) | 74.29 | 19.90 | 78.65 | 17.50 | 78.72 | 21.35 | 83.41 | 15.09 |
| | KGP (Wang et al., 2024b) | 57.51 | 27.34 | 53.51 | 26.59 | 59.38 | 56.20 | 68.42 | 43.85 |
| | StructRAG (Li et al., 2024) | 63.25 | 37.26 | 61.75 | 35.68 | 62.55 | 32.01 | 62.76 | 46.75 |
| | KET-RAG (Huang et al., 2025) | 86.44 | 57.07 | 80.62 | 30.86 | 89.07 | 44.59 | 44.06 | 32.38 |

## C.3 EVIDENCE COLLECTION

Real-world reasoning complexity arises not only from the number of "hops" but from the structural and semantic properties of the knowledge being traversed. Isolating evidence into localized subgraphs (dense concept clusters) and multi-hop chains (logically linked sequences) allows us to quantify difficulty through objective metrics like entity density, path length, and inferential distance. For instance, a subgraph with high entity density tests a model's ability to filter relevant facts within a noisy context, while a long-chain dependency tests its capacity to maintain coherence across logical steps. This stage ensures that "difficulty" is not arbitrarily defined but rooted in the ontology's verifiable properties, aligning question design with the cognitive demands of real analytical tasks. Crucially, this evidence extraction phase distinguishes our approach by ensuring that even simple retrieval questions are anchored in contextually rich subgraphs, while complex reasoning tasks demand traversal of interconnected chains that reflect real-world problem-solving, such as diagnosing a patient by integrating symptoms, lab results, and comorbidities.

## C.4 QUESTION GENERATION

Questions are generated by aligning their cognitive demands with the structural properties of the underlying evidence. Retrieval-focused questions target isolated subgraphs, requiring models to recall clustered facts. Reasoning questions leverage short-range chains, demanding interpretation of relational predicates (e.g., causality or contraindication). Summarization tasks synthesize disjointed subgraphs into narratives, while creation questions extrapolate hypotheses from global graph topology (e.g., predicting policy impacts by traversing regulatory, economic, and clinical subgraphs). Difficulty is calibrated by the depth of contextual synthesis required—retrieval relies on localized subgraphs, whereas creation necessitates integrating hierarchical, relational, and topological cues. The ontology's explicit logic ensures question complexity scales with measurable graph properties, like inferential distance between chain endpoints, avoiding the ambiguity of hop-count-based metrics.

## C.5 CHECK & CORRECT

To ensure the accuracy of both evidence and answers, we perform verification and correction. We define the evidence validation process as follows: given the original corpus and the constructed evidence, we assess whether the evidence can be logically derived from the corpus. Our validation criteria are strict: given a question, if no evidence triple can be inferred from the corpus, the corresponding question is discarded. Similarly, for answer correction, we check whether the provided

***Question Generation: Fact Retrieval***

**-Instructions:**
You are a professional dataset constructor. You can generate questions based on following requirement.
Please generate {num} factual English questions based on the following rules:
**-Generation Guidelines:**
1.Focus on explicit information:
- Character relationships (e.g. family ties, romantic involvement)
- Key events timelines
- Physical descriptions
- Professional roles/occupations
- Geographical locations
2.basic rules:
- each question has a unique and accurate answer
- Cover all characters and core relationships
- Avoid redundancy in questions
3.Prohibited content:
- Subjective interpretations
- Imp lied/non-explicit information

***Question Generation: Complex Reasoning***

**-Task Description:**
You are an expert in constructing reasoning-based datasets from narrative subgraphs. Your task is to generate {num} high-quality English reasoning questions from a given context made up of multiple logical chains. Each question must span multiple triplets and require reasoning across distant elements within a single chain or across multiple chains.
**-Generation Guidelines:**
1. You must generate type2 reasoning questions:
  - Each question must require **multi-hop reasoning** across **at least three triplets**.
  - Preferably, the relevant triplets should be **non-adjacent** (i.e., have some logical distance between them).
  - You may either use a **single chain** or combine **triplets from multiple chains**, as long as the reasoning is clear and explicit.
2. The answer must be uniquely supported by the input triplets.
3. Do not make up or infer anything that is not explicitly stated in the triplets.

***Question Generation: Contextual Summarize***

**-Task Description:**
You are a professional dataset constructor. Your task is to generate summarization-style questions based on the following instructions.
Please generate {num} high-level English questions that summarize or synthesize the information from the provided logical chains.
**-Generation Guidelines:**
1. Questions must focus on **summarizing patterns, events, or roles** across the logical chain.
2. Each question should ideally cover **the majority of the triplets** in a chain, not just isolated facts.
3. You may combine information from **a single chain** or **compare across chains** when appropriate.
5. Avoid factual recall (like who did X to Y); instead, emphasize **storyline progression**, **role dynamics**, or **cause-effect chains**.
6. Do NOT infer or invent unstated details. Use only **explicit** triplets.
7. Do NOT generate questions semantically similar to the existing list.

***Question Generation: Creative Generation***

**-Task Description:**
You are a professional dataset constructor working on a novel understanding benchmark. You are tasked with generating *creative expression questions*, which require imaginative re-narration grounded in factual evidence.
Please generate {num} creative-style questions in the following categories:
- Role-playing narration (first-person): the character retells an event from their point of view
- Textual rewriting (in diary or news report format): the event is re-expressed in another writing style

**-Generation Guidelines:**
1. Role-playing narration questions:
  - Ask the user to imagine being a character and retelling or expressing an experience
  - Use explicit entities and events from the triplets
  - Ground the narrative in observed facts
2. Text rewriting questions:
  - Ask for retelling an event in diary form (subjective), or news form (objective)
  - Diary entries should reflect the character's perspective
  - News reports should be impersonal but factually accurate
3. Each question must be supported by **at least 5 distinct triplets** as evidence. The generated question should synthesize and depend on a *broad scope of factual context*. Do not generate shallow or narrowly grounded questions.

Figure 8: Example prompts used for constructing the Novel Dataset in GraphRAG-Bench.

Table 11: Results of Generation Evaluation using Qwen2.5-14B, covering tasks of varying complexity.

| Category | Model | Fact Retrieval | | Complex Reasoning | | Contextual Summarize | | Creative Generation | | |
|---|---|---|---|---|---|---|---|---|---|---|
| | | ACC | ROUGE-L | ACC | ROUGE-L | ACC | Cov | ACC | FS | Cov |
| *Novel Dataset* | | | | | | | | | | |
| Basic RAG | RAG (w rerank) | 46.74 | 19.11 | 42.36 | 11.90 | 51.55 | 83.00 | 38.23 | 52.27 | 38.76 |
| Graph RAG | MS-GraphRAG (local) (Edge et al., 2024) | 39.89 | 25.93 | 46.12 | 27.70 | 65.28 | 69.10 | 39.57 | 54.64 | 32.42 |
| | MS-GraphRAG (global) (Edge et al., 2024) | 39.89 | 11.54 | 42.22 | 19.10 | 60.41 | 75.45 | 36.59 | 84.60 | 34.30 |
| | HippoRAG2 (Gutiérrez et al., 2025) | 54.79 | 30.16 | 50.45 | 30.30 | 61.14 | 68.99 | 40.52 | 52.24 | 32.05 |
| | LightRAG (Guo et al., 2024) | 44.00 | 12.22 | 40.27 | 9.91 | 52.07 | 86.20 | 39.74 | 78.73 | 39.67 |
| | Fast-GraphRAG (CircleMind-AI, 2024) | 60.08 | 41.31 | 53.81 | 30.43 | 62.82 | 74.45 | 47.60 | 57.99 | 30.31 |
| | RAPTOR (Sarthi et al., 2024) | 41.12 | 11.91 | 41.15 | 9.58 | 52.03 | 81.04 | 38.56 | 63.10 | 34.77 |
| *Medical Dataset* | | | | | | | | | | |
| Basic RAG | RAG (w/ rerank) | 57.56 | 21.14 | 56.01 | 12.71 | 61.95 | 79.32 | 60.91 | 53.84 | 47.91 |
| GraphRAG | MS-GraphRAG (local) (Edge et al., 2024) | 49.24 | 29.17 | 61.64 | 27.40 | 59.01 | 37.12 | 61.70 | 33.39 | 42.24 |
| | MS-GraphRAG (global) (Edge et al., 2024) | 40.04 | 17.01 | 61.40 | 21.08 | 51.10 | 38.69 | 59.20 | - | 43.17 |
| | HippoRAG2 (Gutiérrez et al., 2025) | 64.50 | 33.92 | 64.05 | 33.02 | 64.71 | 48.27 | 60.77 | 39.33 | 36.76 |
| | LightRAG (Guo et al., 2024) | 64.43 | 40.37 | 64.66 | 28.26 | 69.37 | 59.81 | 63.25 | 70.84 | 45.12 |
| | Fast-GraphRAG (CircleMind-AI, 2024) | 62.03 | 43.59 | 62.40 | 29.42 | 65.09 | 46.29 | 62.98 | 31.98 | 35.83 |
| | RAPTOR (Sarthi et al., 2024) | 50.48 | 13.49 | 52.86 | 12.96 | 58.94 | 78.63 | 61.45 | 49.91 | 54.46 |

*Due to time and resource constraints, we tested our benchmark on a representative set of GraphRAG models.

answer can be logically inferred from the evidence. This verification and correction process is supported by advanced models combined with human checking for final confirmation.

## C.6    REFINEMENT

We observe that some generated questions may be overly direct, lacking sufficient contextual information, which may affect effective retrieval. To address this, we further enrich and refine the question by incorporating relevant background knowledge. Specifically, for each question, we locate the corpus segment from which the evidence was derived and employ GPT-4.1 to refine the original question by integrating this segment. This ensures that each question not only retains the necessary logical structure but also provides relevant background context, thereby enhancing the overall clarity and rationality of the question.

## D    EXTENDING GRAPHRAG-BENCH TO NEW DOMAINS

In this paper, we used a medical dataset to represent domain knowledge in the initial version of GraphRAG-Bench. The method we constructed the data can be used to extend the benchmark into other fields, such as law and finance. For researchers who wish to add other domains in future work, we provide some suggested methods.

For the legal domain, we offer the following suggestions: 1) CORPUS SELECTION: We suggest collecting data from the following sources: a) EU Case Law which contains 29.8K EU court decisions, mainly from the Court of Justice (CJEU), published in EUR-Lex. b) UK Case Law which contains 47K UK court decisions from the British and Irish Legal Information Institute (BAILII) database. c) US Case Law which contains 4.6M US decisions (opinions) from Court Listener, a web database hosted by the Free Law Project. 2) DATA MINING METHOD: We recommend a method that integrates both ontology and logic chains. Ontology provides a structured representation of legal regulations and judicial interpretations by defining entities, actions, relationships, and the conditions under which specific laws apply. Logic chains, in parallel, capture the legal reasoning process and model potential multi-hop relationships between different laws.

For the finance domain, we suggest the following: 1) CORPUS SELECTION: We recommend focusing on publicly traded companies from the S&P 500 list for the period between 2015 and 2024. For these companies, we can collect their annual reports, quarterly reports, and reports on unscheduled events from the same period. These documents are primarily available in the EDGAR database. 2) DATA MINING METHOD: We suggest using an ontology to map company information, such as business structures and financial metrics, to a unified schema. It is worth noting that the financial domain contains a large amount of numerical data, which may require further processing. The details statistics of Legal and Financial Corpora are in Table 12.

Table 12: Statistics of Legal and Financial Corpora.

| Source | Numbers | Tokens |
|---|---|---|
| EU Case Law | 14.9K | 89.25M |
| UK Case Law | 11.7K | 92.10M |
| U.S. Case Law | 46.1K | 114.23M |
| **Total** | **72.7K** | **295.58M** |

(a) Statistics of Legal Corpora.

| Report Type | Numbers | Tokens |
|---|---|---|
| Annual Report | 192 | 18.72M |
| Quarterly Report | 588 | 22.93M |
| Current Report | 1427 | 74.21M |
| **Total** | **2207** | **115.86M** |

(b) Statistics of Financial Corpora.

Table 13: Corpus statistics for different benchmarks. We report the average number of entities and relations per 1k tokens (**Avg. Entities, Avg. Relations**), the proportion of non-isolated entities (**Prop. of Non-isolated Entities**), the average node degree (**Avg. Degree**), the proportion of entities with total degree greater than 1, 2, and 3 (**Prop. Degree > $k$**), and the geometric mean of the number of entities in connected components (**Avg. Component Size**).

| Benchmark | Avg. Entities | Avg. Relations | Prop. of Non-isolated Entities | Avg. Degree | Prop. Degree > 1 | Prop. Degree > 2 | Prop. Degree > 3 | Avg. Component Size |
|---|---|---|---|---|---|---|---|---|
| UltraDomain | 170.6 | 73.2 | 0.40 | 0.86 | 0.27 | 0.15 | 0.09 | 2.71 |
| MultiHop-RAG | 10.1 | 3.82 | 0.41 | 0.76 | 0.26 | 0.14 | 0.09 | 2.70 |
| HotpotQA | 39.3 | 12.7 | 0.41 | 0.65 | 0.23 | 0.12 | 0.06 | 2.11 |
| MuSiQue | 21.5 | 6.5 | 0.39 | 0.60 | 0.23 | 0.10 | 0.06 | 2.33 |
| 2WikiMultihopQA | 41.9 | 13.4 | 0.40 | 0.64 | 0.23 | 0.11 | 0.07 | 2.09 |
| GraphRAG-Bench (novel) | 19.6 | 20.9 | 0.66 | 2.27 | 0.47 | 0.25 | 0.17 | 3.99 |
| GraphRAG-Bench (medical) | 11.8 | 6.2 | 0.48 | 1.05 | 0.36 | 0.20 | 0.12 | 3.15 |

# E  DATASET STATISTICS AND VISUALIZATION

**Corpus statistics** We first conduct a statistical analysis of the composition of the corpus from existing benchmarks, as shown in Table 13. We find that these benchmarks contain a large number of redundant entities and relations. Although some benchmarks like UltraDomain have relatively high average numbers of entities (170.6) and relations (73.2), their average degree remains low (with a maximum of only 0.86), indicating a lack of effective connectivity among entities. Further analysis reveals that the average component size remains small (e.g., 2.7 in UltraDomain and MultiHop-RAG), and the proportion of non-isolated entities is also low (around 40%), suggesting sparse and fragmented graph structures. Additionally, only a small fraction of entities have degrees greater than 3 (e.g., 9% in UltraDomain), reflecting limited semantic aggregation. Such structural characteristics lead to low overall

Table 14: Comparison between GraphRAG-Bench and other benchmarks. High Info Density indicates whether the corpus has high information density; Question Diversity denotes whether questions are categorized by difficulty levels; and Reference Answers indicates whether reference answers are provided.

| Dataset | #Questions | High Info Density | Question Diversity | Reference Answers |
|---|---|---|---|---|
| UltraDomain | 2500 | ✗ | ✗ | ✗ |
| Multihop-RAG | 2556 | ✗ | ✗ | ✓ |
| HotpotQA | 7405 | ✗ | ✗ | ✓ |
| GraphRAG-Bench | 4076 | ✓ | ✓ | ✓ |

Figure 9: Distribution of questions with varying difficulty levels across different benchmarks.

information density, making it difficult to effectively support retrieval tasks based on graph structures. These findings highlight the limitations of current benchmarks in terms of information organization and provide a strong motivation for our benchmark design.

Based on the above analysis, GraphRAG-Bench is designed to provide benchmarks with richer and more structured entity-relation graphs than existing datasets. In both the medical and novel subsets, the corpus contains higher average degrees (1.05 and 2.27, respectively), larger proportions of non-isolated entities (0.48 and 0.66), and increased average component sizes (3.15 and 3.99),

indicating more coherent and connected graph structures. These improvements address the sparsity and fragmentation observed in prior benchmarks, and offer a more suitable foundation for studying the impact of graph-based information organization in retrieval-augmented generation. This setup enables controlled experiments for analyzing how different structural properties affect the behavior and effectiveness of GraphRAG systems.

**Questions statistics** We first categorize questions for existing benchmarks based on a predefined taxonomy of difficulty levels, with classification results shown in Figure 9. Our analysis reveals a significant imbalance in the distribution of question levels across current benchmarks. Specifically, UltraDomain focuses mostly on Contextual Summarize questions (97%), while HotpotQA mainly contains Fact Retrieval questions (78.2%), lacking coverage of deeper logical reasoning tasks. Although MultiHop-RAG balances Complex Reasoning and Contextual Summarize questions better, it entirely lacks basic Fact Retrieval questions, making its evaluation coverage incomplete.

To address these limitations, GraphRAG-Bench introduces a carefully designed taxonomy of question types to achieve more comprehensive evaluation coverage. We not only ensure a more balanced distribution across the four core categories but also introduce a novel Creative Generation category, filling a critical gap in assessing generative creativity which largely overlooked by existing benchmarks. This multi-level question design enables GraphRAG-Bench to provide a more systematic evaluation of both RAG and GraphRAG systems, offering unique advantages for analyzing model performance across tasks with varying levels of cognitive

## F    EVALUATION METRICS

**Retrieval Performance**    To evaluate the retrieval performance of GraphRAG, we argue that an effective system should both guarantee the completeness of retrieved information and reduce irrelevant content. We introduce two corresponding retrieval-quality-based metrics as detailed below

• **CONTEXT RELEVANCE** measures how well the retrieved content aligns with the question's intent. It quantifies the semantic similarity between the question and the retrieved evidence, with higher values indicating more focused and pertinent information. Specifically, it can be defined as:

$$\text{CONTEXT RELEVANCE} = \frac{1}{|\mathcal{C}|} \sum_{c \in \mathcal{C}} \text{R}(c, Q, \mathcal{E}), \tag{3}$$

where $\mathcal{C}$ denotes the set of retrieved contexts, $Q$ represents the question, $\mathcal{E}$ denotes the set of evidence, and the operator $\text{R}(\cdot)$ determines whether a context $c$ is relevant to the question $Q$ and the evidence $\mathcal{E}$.

• **EVIDENCE RECALL** measures retrieval completeness by assessing whether all critical components required to correctly answer a question are captured. Higher values indicate more comprehensive evidence collection. The formal definition is as follows:

$$\text{EVIDENCE RECALL} = \frac{1}{|\mathcal{R}|} \sum_{c \in \mathcal{R}} \mathbb{1}\left(\text{S}\left(c, \mathcal{C}\right)\right), \tag{4}$$

where $\mathcal{R}$ is the set of reference claims, and the operator $\text{S}(\cdot)$ determines whether a claim $c$ is supported by the retrieved context $\mathcal{C}$, providing the condition for the indicator function $\mathbb{1}(\cdot)$.

**Generation Accuracy.**    After retrieval, a GraphRAG system is expected to generate accurate answers based on the retrieved contexts. To evaluate the quality of the generation, we introduce four key metrics: 1) LEXICAL OVERLAP: Measures word-level similarity between the generated and reference answers using longest common subsequence matching. 2) ANSWER ACCURACY: Assesses both semantic similarity and factual consistency with the reference answer. 3) FAITHFULNESS: Evaluates whether the relevant knowledge points in a long-form answer are faithful to the given context. 4) EVIDENCE COVERAGE: Measures whether the answer adequately covers all knowledge relevant to the question.

• **ROUGE-L** quantifies text similarity through n-gram overlap between generated and reference answers, capturing both syntactic and semantic alignment (Lin, 2004).

• **ANSWER ACCURACY** provides a dual assessment of answer quality: 1) Semantic alignment: Embedding-based similarity scores 2) Factual precision: Fine-grained statement-level verification

This combined approach ensures answers are both contextually appropriate and factually accurate.

$$AC = \alpha \cdot FC + (1 - \alpha) \cdot SS \tag{5}$$

where $\alpha$ is the weight parameter, we set it to 0.75 by default. FC is the Factual correctness and SS is Semantic similarity, they are defined as:

$$\begin{cases} FC = 2 \cdot \dfrac{\text{TP}}{\text{TP} + \text{FP} + \text{FN}}, \\ SS = \cos(\mathbf{f}_i, \mathbf{c}_j), \end{cases} \tag{6}$$

• **FAITHFULNESS** specifically targets hallucination risks by measuring claim-to-evidence alignment. The metric calculates what percentage of answer assertions are directly supported by the retrieved context, crucial for evaluating retrieval-grounded generation. The computation follows:

$$\text{FS} = \frac{|\{c \in A \mid S(c, C)\}|}{|A|} \tag{7}$$

where FS is the faithfulness score, $A$ is the set of claims in the response, $C$ is the retrieved context, and $S(c, C)$ is a boolean function indicating whether claim $c$ is supported by context $C$.

• **EVIDENCE COVERAGE** evaluates answer completeness against reference standards. Rather than simple overlap, it assesses whether all necessary knowledge components appear in the generated answer, particularly important for complex queries requiring comprehensive responses. The formal computation is as follows

$$\text{Cov} = \frac{|\{e \in E \mid M(e, G)\}|}{|E|} \tag{8}$$

where Cov is the coverage score, $E$ is the set of reference evidences, $G$ is the generated answer, and $M(e, G)$ is a boolean function indicating whether evidence $e$ appears in $G$.

# G    ADDITIONAL EXPERIMENTS

## G.1    EXPERIMENTS ON MORE GRAPHRAG FRAMEWORKS

We evaluated a total of 11 GraphRAG frameworks in our study. Due to space constraints in the main paper, only a subset of the results is presented there. This appendix provides the complete and detailed results for all frameworks, which are shown in Table 9 and Table 10.

## G.2    EXPERIMENTS ON OPEN-SOURCE MODEL

In our main experiments, we employ GPT-4o-mini as the default backbone model for generation. To evaluate how well different GraphRAG frameworks adapt across generation models, we use Qwen2.5-14B as the open-source model. The experimental results are presented in Table 11. Due to time and resource constraints, we tested a representative subset of these models on Qwen2.5-14B. We summarize the observations as follows.

When integrated with the open-source Qwen2.5-14B model, several lightweight GraphRAG frameworks exhibit competitive performance. Notably, Fast-GraphRAG achieves the highest accuracy (60.08%) and ROUGE-L (41.31%) in fact retrieval, as well as strong performance in creative generation (ACC 47.60%) on the Novel dataset. On the Medical dataset, LightRAG leads in Contextual Summarize (ACC 69.37%) and creative generation (FS 70.84%), while HippoRAG2 obtains the best ROUGE-L scores for both fact retrieval (33.92%) and complex reasoning (33.02%). These results suggest that even under open-source settings, resource-efficient GraphRAG frameworks can effectively leverage graph-structured context to support various generation tasks.

## G.3    EXPERIMENTS ON GRAPH CONSTRUCTION EFFICIENCY

To assess the practical deployment costs of different frameworks, we evaluated the graph construction efficiency on the Novel dataset. We used one book (approximately 56k tokens) as the input corpus

Table 15: Construction time and all token usage during the indexing phase across different GraphRAG.

| Method | Time (s) | Input Tokens | Completion Tokens | Total Tokens |
|---|---|---|---|---|
| MS-GraphRAG (Edge et al., 2024) | 292.45 | 535832 | 118841 | 654673 |
| LightRAG (Guo et al., 2024) | 710.32 | 403401 | 70771 | 474172 |
| Fast-GraphRAG | 281.74 | 187765 | 64052 | 251817 |
| HippoRAG (Gutiérrez et al., 2024) | 77.42 | 154990 | 22971 | 177961 |
| HippoRAG2 (Gutiérrez et al., 2025) | 96.71 | 283498 | 46495 | 329993 |
| KGP (Wang et al., 2024b) | 32.01 | 68540 | 20675 | 89215 |
| RAPTOR (Sarthi et al., 2024) | 135.21 | 113641 | 1900 | 115541 |
| KET-RAG (Huang et al., 2025) | 350.43 | 433191 | 84360 | 517551 |
| LAZY-GraphRAG | 253.59 | 519394 | 71756 | 591150 |

and recorded the construction time, total input tokens, and completion tokens during the indexing phase. The quantitative results are presented in Table 15.

We summarize the observations as follows. The efficiency varies significantly across different GraphRAG methods. Notably, KGP demonstrates the highest efficiency, completing the construction process in just 32.01 seconds with the lowest total token usage (89,215). HippoRAG and HippoRAG2 also perform efficiently, maintaining low time and token costs. In contrast, LightRAG and the MS-GraphRAG incur much higher resource consumption. LightRAG requires over 710 seconds to index, while MS-GraphRAG consumes nearly 655,000 tokens. These results suggest that lightweight frameworks offer a far more cost-effective solution for graph indexing, which is critical for applications with limited time or computational budgets.

Table 16: Generation Evaluation (ACC) on Medical dataset across Qwen series models (3B-14B)

| Model | Fact Retrieval | Complex Reasoning | Contextual Summarize | Creative Generation | Avg |
|---|---|---|---|---|---|
| *GraphRAG (HippoRAG2)* | | | | | |
| Qwen2.5-3b | 60.19 | 58.11 | 58.69 | 51.34 | 57.08 |
| Qwen2.5-7b | 65.65 | 64.25 | 64.26 | 51.85 | 61.50 |
| Qwen2.5-14b | 65.98 | 62.62 | 64.95 | 62.09 | 63.91 |
| *RAG* | | | | | |
| Qwen2.5-3b | 58.66 | 52.25 | 58.71 | 54.89 | 56.13 |
| Qwen2.5-7b | 57.45 | 53.53 | 60.52 | 55.65 | 56.79 |
| Qwen2.5-14b | 57.56 | 56.01 | 61.95 | 60.91 | 59.10 |

## G.4    EXPERIMENTS ON THE IMPACT OF LLM SIZE

To investigate the dependence of GraphRAG performance on the underlying LLM capability, we conducted experiments using the Qwen2.5 series (3B, 7B, and 14B) on the Medical dataset. We compared Standard RAG against GraphRAG, employing HippoRAG2 as the representative GraphRAG method. Since the graph index construction is an offline process, we utilized the same pre-constructed index (generated by GPT-4o-mini) used in our main experiments across all model sizes. This experimental design allows us to isolate the impact of model size specifically during the online generation phase. The quantitative results are presented in Table 16.

GraphRAG exhibits a higher sensitivity to model capacity than Standard RAG. While RAG's performance remains relatively flat across scales (from Avg 56.13 to 59.10), GraphRAG demonstrates substantial growth (from Avg 57.08 to 63.91), indicating a greater reliance on the model's reasoning ability to synthesize structural information. Notably, we observe a distinct performance inflection point at the 7B parameter scale (Avg increasing from 57.59 to 61.50), suggesting that 7B serves as a practical "minimum size" threshold where the model acquires sufficient reasoning power to effectively leverage graph-based context.

## G.5    EXPERIMENTS ON SCALABILITY WITH CORPUS SIZE

To evaluate the scalability of GraphRAG compared to Standard RAG across varying data volumes, we conducted controlled experiments on the Novel dataset. We defined the corpus sizes based on token counts calculated via tiktoken: Small (∼56k tokens, representing a single book unit), Medium

Table 17: Generation Evaluation (ACC) across Novel dataset under different corpus Sizes

| Corpus Type | Corpus Size | Fact Retrieval | Complex Reasoning | Contextual Summarize | Creative Generation |
|---|---|---|---|---|---|
| *GraphRAG (HippoRAG2)* | | | | | |
| Small | 56k | 60.14 | 53.38 | 64.10 | 48.28 |
| Medium | 603k | 59.99 | 56.67 | 65.77 | 50.06 |
| Large | 1132k | 59.19 | 54.29 | 62.63 | 51.18 |
| *RAG* | | | | | |
| Small | 56k | 64.73 | 58.64 | 65.75 | 60.61 |
| Medium | 603k | 58.43 | 41.33 | 62.06 | 52.54 |
| Large | 1132k | 58.04 | 43.20 | 62.43 | 47.19 |

(∼603k tokens), and Large (∼1132k tokens, representing the full dataset). We utilized HippoRAG2 as the representative GraphRAG method and compared it against Standard RAG across these scales. The quantitative results are presented in Table 17.

The comparison reveals a critical insight regarding robustness. Standard RAG suffers significant performance degradation as the corpus size increases. Notably, its accuracy in Complex Reasoning drops heavily from 58.64% (Small) to 43.20% (Large). This supports the hypothesis that vector-based retrieval is prone to capturing high-similarity but irrelevant noise as the search space expands. In contrast, GraphRAG (HippoRAG2) exhibits remarkable stability, maintaining consistent performance across all scales (e.g., Fact Retrieval remains steady at ∼60%). This confirms that the performance stability is driven by the structural constraints of the graph (i.e., explicit entity and triple matching), which effectively filter out the retrieval noise that tends to accumulate with increasing corpus size.

# H  IMPLEMENTATION DETAILS

## H.1  IMPLEMENTATION DETAILS OF REPRESENTATIVE RAG MODELS

Table 18: Implementation Details of different GraphRAG models.

| Model | Indexing | | Retrieval | | Generate |
|---|---|---|---|---|---|
| | Knowledge Type | Index Content | Query Input | Granularity | LLM context |
| RAG | Plain Text | Text Chunk | Query Embedding | Chunk | Literal Text |
| MS-GraphRAG(local) (Edge et al., 2024) | Textual Knowledge Graph | Entity,Community | Query Embedding | Entity,Relationship,Chunk,Community | Tabular Result |
| MS-GraphRAG(global) (Edge et al., 2024) | Textual Knowledge Graph | Community | Query Embedding | Community(Layer) | Literal Text |
| HippoRAG (Gutiérrez et al., 2024) | Knowledge Graph | Entity | Entities in Query | Chunk | Reasoning path |
| HippoRAG2 (Gutiérrez et al., 2025) | Knowledge Graph | Phrase,Passage | Query Embedding | Phrase, Chunk | Literal Text |
| LightRAG (Guo et al., 2024) | Textual Knowledge Graph | Entity,Relationship | Keywords in Query | Entity,Relationship,Chunk | Literal Text + Graph Element |
| FastGraphRAG (CircleMind-AI, 2024) | Textual Knowledge Graph | Entity | Entities in Query | Entity,Relationship,Chunk | Literal Text |
| RAPTOR (Sarthi et al., 2024) | Tree | Treenode | Query Embedding | Tree node | Reasoning path |
| KGP (Wang et al., 2024b) | Knowledge Graph | Entity, Relationship | Query Graph | Subgraph | Linearized Subgraph |
| LAZY-GRAPHRAG | Plain Text (on-demand graph) | Text Chunk | Query Embedding | Chunk, Node (dynamic) | Literal Text |
| structRAG (Li et al., 2024) | Hierarchical Knowledge Graph | Hierarchical Node | Query Embedding | Node, Path | Literal Text + Structural Info |
| KET-RAG (Huang et al., 2025) | Knowledge-Enhanced Tree | Keyword, Entity, Chunk | Query (multi-index) | Keyword, Entity, Chunk | Literal Text |

As discussed in previous work (Gao et al., 2023; Lewis et al., 2020; Zhou et al., 2025), RAG models comprise three core components: indexing, retrieval, and generation, each with its specific implementation details as presented in Table 18. Some explanation should be given to the content in the "Knowledge Type" column of the table.A knowledge graph is constructed by extracting entities and relationships from each chunk, which contains only entities and relations, is commonly represented as triples. A textual knowledge graph is a specialized KG (following the same construction step as knowledge graph), which enriches entities with detailed descriptions and type information. A tree structure formed by document content and summary.

**Implementation Details of RAG** We follow the standard RAG paradigm: a retriever model first retrieves relevant context from the corpus based on the given question, and then the question is concatenated with the retrieved context to form a query for the generation model to produce the final answer. Since existing RAG approaches often incorporate rerankers to improve retrieval quality, we consider two baselines: RAG-with-rerank and RAG-without-rerank.

**Implementation Details of GraphRAG** We evaluate several representative GraphRAG frameworks on our benchmark, including:

- MS-GRAPHRAG(LOCAL): Microsoft-GraphRAG based on local retrieval granularity.

- MS-GRAPHRAG(GLOBAL): Microsoft-GraphRAG based on global retrieval granularity.

- LIGHTRAG: a framework that enhances graph efficiency by leveraging optimized graph structures and a two-stage retrieval pipeline.

- HIPPORAG: a framework inspired by the hippocampal memory indexing theory, integrating large language models, knowledge graphs, and personalized PageRank to enable efficient single-step multi-hop knowledge integration and retrieval.

- HIPPORAG2: a framework that achieves deeper contextual understanding by jointly incorporating conceptual (phrase-level) and contextual (passage-level) nodes.

- FAST-GRAPHRAG: a framework designed to improve retrieval speed and reduce computational cost through efficient graph-based querying.

- RAPTOR: a framework that constructs a tree structure through recursive embedding, clustering, and summarization of text segments, enabling efficient information retrieval across different levels of abstraction.

- STRUCTRAG: a framework that boosts knowledge-intensive reasoning of LLMs by dynamically restructuring scattered information into a hybrid, structured format at inference time.

- KGP: a framework that improves multi-document question answering by constructing and traversing a knowledge graph to formulate the right context for large language models.

- LAZY-GRAPHRAG: a framework that achieves a better cost-quality trade-off by using a "lazy" approach to build a concept graph only at query time.

- KET-RAG: a framework that achieves a cost-efficient and high-quality Graph-RAG by leveraging a multi-granular indexing approach combining a knowledge graph skeleton with a text-keyword bipartite graph.

All of these GraphRAG methods construct graphs and refine retrieval strategies to boost RAG system performance across various specialized tasks. Due to time limits, we only assess several representative GraphRAG models. We will include more SOTA models, like ArchRAG (Wang et al., 2025b) PIKE-RAG (Wang et al., 2025a) MedRAG (Zhao et al., 2025) PathRAG (Chen et al., 2025a) DBCopilot (Wang et al., 2025c) LightPROF (Ao et al., 2025) CG-RAG (Hu et al., 2025).

## H.2 CONFIGURATION OF GRAPHRAG MODELS

In our experiments, we maintained consistent conditions for fair comparison. Specifically, all GraphRAG and RAG systems used the bge-large-en-v1.5 embedding model during retrieval stage, and used a generation temperature of 0.7 during generation stage. For GraphRAG systems, given the inherent differences in graph indexing, retrieval strategies, and generation mechanisms across frameworks, we preserved their default configurations (including graph indexing, retrieval strategies, and generation methods) without modification to assess their native performance. This approach ensures both comparability across systems and realistic evaluation of their practical capabilities. The detailed configuration parameters are following:

**RAG Configuration**

```
{
  "embedding_model": "bge-large-en-v1.5",
  "reranker": "bge-reranker-large",
  "retrieval_topk": 5
  "chunk_token_size": 256,
}
```

### MS-GraphRAG(global&local) Configuration

```
{
  "embedding_model": "bge-large-en-v1.5",
  "chunk_token_size": 1000,
  "chunk_overlap_token_size": 100,
  "summarize_descriptions_max_length": 500,
  "max_cluster_size": 10,
  "community_reports_max_length": 2000,
  "community_reports_max_input_length": 8000
}
```

### LightRAG Configuration

```
{
  "embedding_model": "bge-large-en-v1.5",
  "query_type": "hybrid",
  "chunk_token_size": 1200,
  "retrieval_topk": 30,
  "chunk_overlap_token_size": 100,
  "max_token_for_text_unit": 4000,
  "max_token_for_global_context": 4000,
  "max_token_for_local_context": 4000
}
```

### FastGraphRAG Configuration

```
{
  "embedding_model": bge-large-en-v1.5,
  "entity_ranking_policy": 0.005,
  "relation_ranking_policy": 64,
  "chunk_ranking_policy": 8,
}
```

### HippoRAG2 Configuration

```
{
  "embedding_model": bge-large-en-v1.5,
  "retrieval_top_k": 5,
  "linking_top_k": 5,
  "max_qa_steps": 3,
  "qa_top_k": 5,
  "graph_type": facts_and_sim_passage_node_unidirectional,
}
```

### HippoRAG Configuration

```
{
  "embedding_model": bge-large-en-v1.5,
  "chunk_token_size": 1200,
  "chunk_overlap_token_size": 100,
  "retrieve_topk": 20,
  "entities_max_tokens": 2000,
  "relationships_max_tokens": 2000,
}
```

---

**RAPTOR Configuration**

```
{
  "embedding_model": bge-large-en-v1.5,
  "chunk_token_size": 1200,
  "chunk_overlap_token_size": 100,
  "num_layers": 5,
  "max_length_in_cluster": 3500,
  "threshold": 0.1,
  'cluster_metric': cosine,
  'threshold_cluster_num': 5000
}
```

# I   RELATED WORK

## I.1   TRADITIONAL RAG AND THEIR LIMITATIONS

The naive RAG systems (Lewis et al., 2020) operate through three key steps: knowledge preparation, retrieval, and integration. During knowledge preparation, external sources such as documents, databases, or webpages are divided into manageable textual chunks and converted into vector representations for efficient indexing. In the retrieval stage, when a user submits a query, the system searches for relevant chunks using keyword matching or vector similarity measures. The integration stage then combines these retrieved chunks with the original query to create an informed prompt for the LLM's response. Recent advancements in RAG systems have moved beyond basic text retrieval to structured knowledge integration Xiao et al. (2026; 2025); Chen et al. (2025b). Modern implementations employ hierarchical architectures maintaining document organization via layered retrieval processes (Chen et al., 2024a; Li et al., 2024), while others enhance precision through multi-phase retrieval mechanisms that first broaden then refine context selection (Glass et al., 2022; Xu et al., 2023). Autonomous query parsing frameworks automatically break down intricate questions into executable subqueries (Asai et al., 2023), complemented by context-aware systems that modify retrieval tactics in real-time according to query complexity and intent (Tang et al., 2024; Sarthi et al., 2024). These strategies advance naive RAG systems by improving context awareness, retrieval accuracy, and handling complex queries.

Although researchers have extensively explored traditional RAG, there are still some unresolved limitations due to the constraints of the data structure itself. (i) Vector database architectures limit traditional RAG's ability to handle multi-hop reasoning, as they retrieve information only from text chunks containing anchor entities. While methods like query expansion (Mao et al., 2020) and metadata enrichment (Wang et al., 2024a) attempt to improve complex query handling, they remain constrained by the chunk-based knowledge structure that inherently disconnects related concepts. This structural limitation particularly hinders domain-specific reasoning requiring logical synthesis across distributed evidence. (ii) The chunking process sacrifices critical contextual relationships between specialized terms and abstract concepts, despite techniques like real-time retrieval alignment (Jiang et al., 2024) and external API integration (Lazaridou et al., 2022). Vector databases' flat organization fails to preserve hierarchical or conceptual dependencies essential for domain expertise utilization, leaving models unable to reconstruct professional knowledge networks from fragmented chunks. (iii) Vector similarity retrieval often overwhelms LLMs' fixed context windows (OpenAI, 2023; Anthropic, 2024) with redundant content, exacerbating their limited capacity for long-range dependency modeling. While strategies like context pruning (Arefeen et al., 2024) and LLM fine-tuning (Luo et al., 2023) reduce input volume, they cannot compensate for the structural inability to establish explicit connections between retrieved chunks. This fundamental mismatch persists despite optimizations like sliding windows (Wang et al., 2024a), as vector-based approaches lack mechanisms for relational reasoning.

## I.2   GRAPHRAG AND ITS ADVANTAGES

To address this, graph retrieval-augmented generation (GraphRAG) (Peng et al., 2024; Procko & Ochoa, 2024) has recently emerged as a powerful paradigm that leverages external structured graphs

***Question Generation: Fact Retrieval***

**-Instructions:**
You are a professional dataset constructor. You can generate questions based on following requirement.
Please generate {num} factual English questions based on the following rules:
**-Generation Guidelines:**
1.Create information-seeking questions that require direct factual retrieval from the relations list.
Focus on these biomedical categories:
- Cancer Type identification- Disease staging
- Biomarker associations
- Treatment protocols
- Diagnostic methods
- Risk factors
- Symptomatology
2.basic rules:
- each question has a unique and accurate answer
- Cover all medical concept,Ensure proportional coverage of all concept categories
-** Avoid redundancy in questions**
3.Prohibited content:
- Subjective interpretations
- Implied/non-explicit information

***Question Generation: Complex Reasoning***

**-Task Description:**
You are a professional dataset constructor. You can generate questions based on following requirement.
Generate {num} Reasoning English questions,each questions requiring multiple inferential leaps or accessing several pieces of
information from different locations or sources to arrive at an answer. Generate based on the following rules:
**-Generation Guidelines:**
Generation Guidelines:
1.Focus on explicit information:
- the relations that have the same entity  - find the connections of three or more evidences and combin them to form a more complex idea.
2.basic rules:
- each question has a unique and accurate answer  - each question cannot be answered by relying on just one evidence but instead requires
understanding and linking the information from different sentences.  - Avoid redundancy in questions
3.Do not make up or infer anything that is not explicitly stated in the input.

***Question Generation: Contextual Summarize***

**-Task Description:**
You are a professional dataset constructor. You can generate questions based on the following requirement.Generate 5-10 **summary-type
English questions**, where each question prompts a concise synthesis of biomedical knowledge related to cancer based on the input data.
**-Generation Guidelines:**
1.Avoid factual recall (like who did X to Y),instead,focus on synthesis of explicit information:
  - Summarize categories, patterns, or properties connected to a single concept (e.g., all diagnostic methods used for NSCLC).
  - Consolidate related items (e.g., all symptoms of a condition, all treatments available at a specific stage)
2.Each question must have one clear, unique answer and should ideally cover **the majority of the ontology**
3.Avoid redundancy and overly narrow questions.
4.Do NOT infer or invent unstated details or use any information not explicitly stated in the input or ontology.

***Question Generation: Creative Generation***

**-Task Description:**
You are a professional dataset constructor working on a novel understanding benchmark. You are tasked with generating *creative
expression questions*, which require imaginative re-narration grounded in factual evidence.
Please generate {num}clinical synthesis tasks in the following categories.
-Role-based medical tasking: the model is asked to take on a professional medical role (e.g., clinician, radiologist, medical intern, patient)
and perform a complex task like diagnosis formulation, treatment planning, or case summary writing.
-Format transformation: rewrites the medical facts into a specific document format such as a clinical report, referral note, discharge
summary, or public health advisory.
**-Generation Guidelines:**
1.Role-based medical tasking:
  - Ask the user to imagine they are a medical professional or patient
  - Task examples: write a diagnosis, generate a treatment plan, construct a referral letter, etc.
  - The response should reflect professional medical tone and reasoning
2.Format transformation:
  - Ask the user to rewrite the case facts into a formal medical document
  - Examples: Discharge summary, SOAP note, progress note, guideline draft, public notice
  - Use explicit entities from the triplets and follow relevant medical document conventions

Figure 10: Example prompts used for constructing the Medical Dataset in GraphRAG-Bench.

to improve LLMs' capability on contextual comprehension (Han et al., 2024; Zhang et al., 2025a). Early efforts, like Microsoft GraphRAG (Edge et al., 2024) and its variant LazyGraphRAG (Darren Edge, 2024), employ hierarchical community-based search and combine local/global querying for comprehensive responses. Building on this, LightRAG (Guo et al., 2024) improves scalability through dual-level retrieval and graph-enhanced indexing, while GRAG (Hu et al., 2024) introduces a soft pruning technique to mitigate the impact of irrelevant entities in retrieved subgraphs and employs graph-aware prompt tuning to help LLMs interpret topological structure. Further extending these capabilities, StructRAG (Li et al., 2024) tailors data structures to specific tasks by dynamically selecting optimal graph schemas, while KAG (Liang et al., 2024) constructs domain expert knowledge using conceptual semantic reasoning and human-annotated schemas, which significantly reduces noise present in OpenIE systems. These strategies used in GraphRAG models significantly improve retrieval precision and contextual depth, enabling LLMs to address complex, multi-hop queries more effectively.

GraphRAG offers several significant advantages over traditional RAG systems (Peng et al., 2024), enhancing the capabilities of AI-powered information retrieval and generation. First, its graph-based knowledge representation captures hierarchical relationships and multi-hop dependencies between entities, enabling nuanced contextual reasoning and discovery of latent connections (Yang et al., 2026). This structured approach resolves ambiguity by evaluating multiple semantic paths during query processing. Besides, the graph structure allows unified integration of structured databases, semi-structured formats, and unstructured text within a single graph, supporting cross-modal queries that combine textual, numerical, and multimedia data (Yin et al., 2023; 2020; Yue et al., 2024). This interoperability maximizes value from heterogeneous organizational knowledge assets (Procko & Ochoa, 2024; Luo et al., 2024; 2025; Xia et al., 2025). Third, users can audit decision pathways by visualizing entity relationships traversed during retrieval. Combined with LLMs, this transparent architecture supports multi-hop logical synthesis, critical for specialized domains like healthcare and finance (Procko & Ochoa, 2024; Han et al., 2024).

### I.3 EXISTING BENCHMARKS AND ANALYSIS

It is crucial to identify the factors that are currently limiting GraphRAG's real-world performance. However, quantitatively and fairly assessing the role of graph structures in RAG systems is challenging. Current benchmarks, including HotpotQA (Yang et al., 2018), MultiHopRAG (Tang & Yang, 2024) and UltraDomain (Qian et al., 2024), fail to adequately evaluate the effectiveness of graph structures in RAG systems due to fundamental limitations in both their problem design and corpus composition. A few studies, like DIGIMON (Zhou et al., 2025) and another analysis paper (Han et al., 2025), have recently tried to analyze the effect of different GraphRAG models. Despite their effort, they mainly focus on architectural comparisons using homogeneous datasets, missing how models synthesize hierarchical expertise and unstructured narratives. To this end, we propose GraphRAG-Bench, a comprehensive benchmark designed to evaluate GraphRAG models on deep reasoning. It features hybrid corpora with tasks of increasing complexity and stage-specific metrics to expose why models fail, whether in graph construction, knowledge retrieval, or contextual synthesis. Leveraging this novel benchmark, we systematically investigate the conditions when GraphRAG surpasses traditional RAG systems and the underlying reasons for its success, offering guidelines for its practical application.

## J LIMITATION

While our benchmark advances GraphRAG evaluation by systematically addressing reasoning complexity beyond traditional retrieval-centric paradigms, it inherits constraints inherent to its design scope. Most notably, the framework operates exclusively within unimodal (text-based) contexts, omitting the challenges and opportunities posed by multimodal data integration. Real-world applications of GraphRAG often necessitate synthesizing heterogeneous information types, such as visual diagrams, tabular datasets, or sensor-generated temporal sequences, to resolve complex queries. This limitation mirrors a broader gap in RAG benchmarking, as existing frameworks similarly neglect multimodal interplay despite its growing practical relevance. Future iterations will expand this work to incorporate multimodal evaluation, testing how graph-based retrieval and reasoning mechanisms generalize to hybrid knowledge representations while preserving contextual fidelity across data types.

## K    BROADER IMPACT

Our work introduces a paradigm shift in how to comprehensively evaluate GraphRAG systems, with broader implications for AI's role in knowledge-intensive domains such as healthcare, legal analysis, and scientific research. By rigorously assessing not only the outputs but also the structural and procedural integrity of knowledge representation and reasoning, our benchmark advances the development of AI systems capable of contextually grounded, logically coherent problem-solving. This progress addresses a critical gap in current AI evaluation methodologies, which often prioritize superficial fluency over semantic and causal fidelity, thereby risking the deployment of systems that generate plausible but ungrounded or fragmented insights.

From a technical perspective, our framework establishes a precedent for holistic evaluation, encouraging the AI community to move beyond answer-centric metrics and instead prioritize the traceability of reasoning processes. This shift could catalyze innovations in graph-based knowledge representation, fostering models that explicitly encode domain hierarchies, causal relationships, and contextual dependencies, capabilities essential for real-world applications like clinical decision support or policy analysis. For instance, by evaluating how faithfully a system traverses medical guideline graphs to synthesize treatment recommendations, our approach incentivizes the development of reliable, domain-aware AI, reducing reliance on opaque black-box reasoning.

## L    THE USE OF LARGE LANGUAGE MODELS (LLMS)

In the preparation of this manuscript, we used a large language model as a writing assistant. Its main role was to help improve our English writing, such as correcting grammar and refining sentences for clarity and style. Additionally, it was used to help set up the initial format for several tables. The authors made all final decisions on the content, carefully checking and editing all suggestions from the model to ensure the scientific accuracy and integrity of this work.

