# OpenReview forum: "When to use Graphs in RAG: A Comprehensive Analysis for Graph Retrieval-Augmented Generation"
_ICLR.cc/2026/Conference — ICLR 2026 Poster_

### Official Review · Reviewer_bvTB · 2025-10-21

**Soundness:** 4
**Presentation:** 4
**Contribution:** 3
**Rating:** 8
**Confidence:** 4

**Summary:**

This paper systematically investigates when and why graph structures benefit retrieval-augmented generation systems, introducing **GraphRAG-Bench** to evaluate GraphRAG models across hierarchical knowledge retrieval and deep contextual reasoning, and providing empirical insights and guidelines for their practical application.

**Strengths:**

1. This paper addresses the key research question of when graph structures truly improve RAG performance.
2. This paper provides extensive experimental evidence through the new GraphRAG-Bench benchmark.
3. This paper offers concrete recommendations and design principles for advancing the GraphRAG community.

**Weaknesses:**

1. **The implementation details of the evaluation metrics are not clearly specified.** It seems that both *Faithfulness* and *Evidence Coverage* rely on external judgment models, such as *LLM-as-a-Judge*, but the paper does not disclose the exact implementation of this component.

2. **The scale of retrieved content among methods appears inconsistent**. According to Section H.2 *Configuration*, the retrieval strategies of RAG and different GraphRAG variants are not aligned—for example, RAG restricts the Top-K to 5. This inconsistency may affect the fairness of performance comparisons.

3. **The potential mechanisms through which *Graph Indexing content* influences various tasks deserve further study**. Statistics in Table 15 show that different GraphRAG strategies employ distinct types of indexed content, such as text chunks, communities, entities, and relationships. For instance, could the inclusion of *community* information better support *contextual summarization* tasks?

4. **How corpus characteristics affect the choice of GraphRAG remains an open question**. Although the paper thoroughly investigates when to use GraphRAG from the perspective of task properties and performance, Sections C.1 and D further discuss the traits of the selected *Medical* and *Novel* corpora, as well as the yet-unexplored *Legal* and *Financial* domains. Extending the study of "When to use GraphRAG" to domain-specific corpora would significantly strengthen the work.

**Questions:**

As discussed above in Weakness.

---

> ### Author Response · Authors · 2025-11-22
>
> Dear reviewer bvTB,
>
> Thanks a lot for your detailed feedback. We really appreciate your recognition of our work and also appreciate your time and effort in providing insightful suggestions that can help further polish our paper. Below are detailed responses to your comments and suggestions:
>
> > W1: **The implementation details of the evaluation metrics are not clearly specified.** It seems that both Faithfulness and Evidence Coverage rely on external judgment models, such as LLM-as-a-Judge, but the paper does not disclose the exact implementation of this component.
>
> Thank you for raising this important point. You are correct that the paper does not include a detailed description of how the Faithfulness and Evidence Coverage metrics are implemented. To address this, we provide the specific implementation below, including the pseudocode and the exact prompts used for the LLM judge.
>
> **The detailed evaluation process for Faithfulness:**
> ```pseudo
> prompt_generate_statements = """
> Given a question and an answer, analyze the complexity of each sentence in the answer. Break down each sentence into one or more fully understandable statements. Ensure that no pronouns are used in any statement. Format the outputs in JSON.
>
> [Few-shot Examples]
>
> Question:{question}
> Answer: {answer}
>
> Your Response:
> """
>
> prompt_evaluate_statements = """
> Your task is to judge the faithfulness of a series of statements based on a given context. For each statement you must return verdict as 1 if the statement can be directly inferred based on the context or 0 if the statement can not be directly inferred based on the context.
>
> [Few-shot Examples]
>
> Context: {context}
> Statements: {statements}
>
> Your Response:
> """
>
> function compute_faithfulness_score(question, answer, contexts, llm):
>     statements ← generate_statements(question, answer, llm)
>     if statements is empty:
>         return 1.0 if answer is empty else NaN
>
>     context_str ← concatenate(contexts)
>     if context_str is empty:
>         return 0.0
>
>     verdicts ← evaluate_statements(statements, context_str, llm)
>     if verdicts is empty:
>         return NaN
>
>     return average(v.verdict for v in verdicts)
>
> ```
>
> **The detailed evaluation process for Evidence Coverage:**
> ```pseudo
> prompt_extract_facts = """
> You are given a question and a reference answer. Break down the reference answer into a list of distinct factual statements (facts) that could be independently verified.
> Output them as a JSON list of strings under the 'facts' field.
>
> [Few-shot Examples]
>
> Question: {question}
> Reference Answer: {reference}
>
> Your Response:
> """
>
> prompt_check_fact_coverage = """
> For each factual statement from the reference, determine if it's covered in the response.
> Respond ONLY with a JSON object containing a "classifications" list. Each item should have:
> - "statement": the exact fact from reference
> - "attributed": 1 if covered, 0 if not
>
> [Few-shot Examples]
>
> Question: {question}
> Response: {response}
> Reference Facts: {facts}
>
> Your Response:
> """
>
> function compute_coverage_score(question, reference, response, llm):
>     if reference is empty:
>         return 1.0
>
>     facts ← extract_facts(reference, llm)
>     if facts is empty:
>         return NaN
>
>     classifications ← check_fact_coverage(question, facts, response, llm)
>     if classifications is empty:
>         return NaN
>
>     return average(c.attributed for c in classifications)
>
> ```
>
>
> Revision：The detailed calculation specifics and prompts for our evaluation metrics are added in **Figure 12-15** of our revised manuscript. This will ensure that our evaluation process is fully transparent and reproducible. Thank you again for pointing this out.

---

> ### Author Response · Authors · 2025-11-22
>
> > W2: **The scale of retrieved content among methods appears inconsistent.** According to Section H.2 Configuration, the retrieval strategies of RAG and different GraphRAG variants are not aligned—for example, RAG restricts the Top-K to 5. This inconsistency may affect the fairness of performance comparisons.
>
> Thank you for raising this valid concern regarding the consistency of retrieved content scales. We acknowledge that the retrieval strategies are not strictly aligned. Due to the fundamentally different mechanisms of GraphRAG variants (e.g., MS-GraphRAG retrieves community reports, while HippoRAG2 traverses graphs to find relevant chunks) and practical constraints on computational cost, it is challenging to enforce a uniform "retrieved token count" or "Top-K" across all diverse frameworks.
>
> To address this, we used a practical approach. For GraphRAG, we followed the best settings recommended in their original papers or codebases. For standard RAG, we chose Top-5. We then tested RAG with different settings (Top-3, Top-5, and Top-8) on the Medical dataset to check the change of performance.
>
> Table 11: The results of generation performance (ACC) of RAG under different Top-k settings.
> | Top-K   | Fact Retrieval | Complex Reasoning  | Contextual Summarize | Creative Generation |
> |---------|------------------|---------------------|-------------------------|------------------------|
> | **top3** | 66.62            | 59.58               | 66.05                   | 60.36                  |
> | **top5** | 64.73            | 58.64               | 65.75                   | 60.61                  |
> | **top8** | 66.59            | 59.96               | 66.75                   | 63.09                  |
>
> As shown in Table 11, the accuracy stays stable with only small changes. Increasing the retrieval to Top-8 does not significantly improve the results. Therefore, we believe Top-5 can be a representative baseline. We appreciate the feedback and will explore better ways to align these settings in the future.
>
> > W3: **The potential mechanisms through which Graph Indexing content influences various tasks deserve further study.** Statistics in Table 15 show that different GraphRAG strategies employ distinct types of indexed content, such as text chunks, communities, entities, and relationships. For instance, could the inclusion of community information better support contextual summarization tasks?
>
> Thank you for this excellent and insightful question. We completely agree that a deeper study into how different indexed graph components (e.g., communities, entities) influence specific task performance is worthwhile. As you mentioned, incorporating community information allows GraphRAG to aggregate information from different text chunks. Our specific analysis is as follows:
>
> Regarding the specific impact of community information on contextual summarization, our results show a clear trend. GraphRAGs that use community structures（specifically MS-GraphRAG and Fast-GraphRAG） achieve the best performance. For example, MS-GraphRAG (local) reached the highest ACC on the Novel dataset (64.40), and Fast-GraphRAG obtained the highest ACC on the Medical dataset (67.88). **This indicates that community information helps group related content, which is directly beneficial for summarization tasks.**
>
> However, we also found this is not a universal solution and introduces a trade-off. **Blindly incorporating community information can lead to the retrieval of irrelevant, redundant content.** As shown in our detailed retrieval results (Tables 4 and 10), both MS-GraphRAG and Fast-GraphRAG exhibit an imbalance of high recall but lower relevance. This indicates that while communities promote information aggregation, it can be at the cost of retrieval precision. Furthermore, we note that strong performance is also achievable without explicit communities. For example, HippoRAG2 also performs very strongly on Contextual Summarization (Novel: 64.10 ACC; Medical: 63.08 ACC) by relying on its efficient indexing and retrieval mechanisms. Therefore, **an effective GraphRAG framework should aim to enhance information aggregation while simultaneously exploring ways to minimize retrieved redundancy and improve retrieval efficiency.**
>
> Thank you for your feedback and suggestions again.

---

> ### Author Response · Authors · 2025-11-22
>
> > W4: **How corpus characteristics affect the choice of GraphRAG remains an open question.** Although the paper thoroughly investigates when to use GraphRAG from the perspective of task properties and performance, Sections C.1 and D further discuss the traits of the selected Medical and Novel corpora, as well as the yet-unexplored Legal and Financial domains. Extending the study of "When to use GraphRAG" to domain-specific corpora would significantly strengthen the work.
>
> Thank you for your valuable feedback. We completely agree that corpus characteristics (such as domain, structure, and density) are a critical factor in determining GraphRAG's effectiveness, and this remains an important open question.
>
> Our current work began this exploration. As noted in Appendix D, our taxonomy categorizes corpora into specialized domains and proprietary data. We selected medical data to represent a specialized, hierarchical domain, and novel data to simulate real-world documents containing implicit, non-linear narratives.
>
> We acknowledge this is a first step, and we are actively planning to extend this research. As you suggests, a key part of our future work will be to construct benchmarks for other complex domains, such as the Legal and Financial domains. This will enable a more systematic investigation into how varying corpus properties impact GraphRAG performance. We appreciate this constructive feedback, as it strongly aligns with our future research goals.

---

### Official Review · Reviewer_ebWh · 2025-10-30

**Soundness:** 4
**Presentation:** 3
**Contribution:** 3
**Rating:** 6
**Confidence:** 4

**Summary:**

This paper proposes GraphRAG-Bench, a comprehensive benchmark for systematically evaluating GraphRAG models in terms of hierarchical knowledge retrieval and deep contextual reasoning.

This benchmark covers both tightly structured domain data (medical data from NCCN guidelines) and loosely organized texts (novels from Gutenberg library), and focuses on 4 tasks with different difficulty levels, from easy to difficult, including fact retrieval, complex reasoning, contextual summarize, creative generation. This paper provides an evaluation framework with clear quantitative metrics that assesses graph construction, retrieval, and generation performance.

This paper conducted extensive experiments to evaluate multiple GraphRAG methods against RAG, with GPT-4o-mini and Qwen-2.5-14B as base LLMs. The experiments show that GraphRAG outperforms RAG in complex reasoning, summarization, and creative tasks, while it does not hold an advantage in simple retrieval. Based on the experimental results, the paper offers practical guidelines, indicating that GraphRAG is suitable for multi-hop reasoning and knowledge integration tasks, while it is not applicable for single-step retrieval or latency-sensitive scenarios.

**Strengths:**

- This benchmark focuses on tasks with different difficulty levels, reflecting real-world scenarios demanding complex logical synthesis.
- This benchmark provides a comprehensive evaluation for GraphRAG with clear quantitative metrics, including graph structure, retrieval performance, efficiency, and final output quality.
- Based on thorough experiments across multiple GraphRAG methods, LLMs, and tasks, this paper offers clear and practical guidelines on when GraphRAG outperforms traditional RAG.

**Weaknesses:**

- The calculation process for some evaluation metrics is simplistic and unclear. For example, the paper does not explain how it determines whether a claim $c$ is supported by the context $C$ when calculating EVIDENCE RECALL and FAITHFULNESS.
- The process of Logic and Evidence Extraction is not clearly described either in the main text or the appendix. While the use of GPT-4.1 is mentioned, the specific extraction procedure and output format are not illustrated clearly.

**Questions:**

Refer to Weaknesses.

---

> ### Author Response · Authors · 2025-11-22
>
> Dear ebWh:
>
> We really appreciate your recognition of our work, and thanks a lot for providing insightful suggestions that can help further polish our paper.
>
> > W1: The calculation process for some evaluation metrics is simplistic and unclear. For example, the paper does not explain how it determines whether a claim is supported by the context when calculating EVIDENCE RECALL and FAITHFULNESS.
>
> Thank you for highlighting this lack of clarity. You are correct that the paper did not sufficiently detail the calculation process, specifically how EVIDENCE RECALL and FAITHFULNESS calculates. These metrics are evaluated using an LLM-as-a-Judge approach. To clarify our implementation, we provide the specific implementation, including the exact prompts and criteria used by theLLM-as-a-Judge to determine if a claim is supported by the given context.
>
> **The detailed evaluation process for EVIDENCE RECALL:**
> ```pseudo
> prompt_get_classifications= """
> You are given a list of evidences and a Context. For each evidence, determine whether it can be attributed to the Context.
>
> Respond ONLY with a JSON object containing a "classifications" list. Each item should include:
> - "statement": the exact evidence string
> - "reason": a brief explanation (1 sentence)
> - "attributed": 1 if the evidence can be attributed to the Context, otherwise 0
>
> [Few-shot Examples]
>
> Context: {context}
>
> Evidence: {evidence}
>
> Question: {question} (for reference only)
>
> Your Response:
> """
>
> function compute_evidence_recall(question, contexts, reference_evidence, llm):
>     context_str ← join(contexts)
>     if context_str is empty:
>         return 0.0
>
>     classifications ← get_classifications(prompt, llm)
>
>     if classifications is empty:
>         return NaN
>
>     return average(c.attributed for c in classifications)
>
> ```
> **The detailed evaluation process for FAITHFULNESS:**
>
> ```pseudo
> prompt_generate_statements = """
> Given a question and an answer, analyze the complexity of each sentence in the answer. Break down each sentence into one or more fully understandable statements. Ensure that no pronouns are used in any statement. Format the outputs in JSON.
>
> [Few-shot Examples]
>
> Question: {question}
> Answer: {answer}
>
> Your Response:
> """
>
> prompt_evaluate_statements = """
> Your task is to judge the faithfulness of a series of statements based on a given context. For each statement you must return verdict as 1 if the statement can be directly inferred based on the context or 0 if the statement can not be directly inferred based on the context.
>
> [Few-shot Examples]
>
> Context: {context}
> Statements: {statements}
>
> Your Response:
> """
>
> function compute_faithfulness_score(question, answer, contexts, llm):
>     statements ← generate_statements(question, answer, llm)
>     if statements is empty:
>         return 1.0 if answer is empty else NaN
>
>     context_str ← concatenate(contexts)
>     if context_str is empty:
>         return 0.0
>
>     verdicts ← evaluate_statements(statements, context_str, llm)
>     if verdicts is empty:
>         return NaN
>
>     return average(v.verdict for v in verdicts)
>
> ```
>
> Revision：The detailed calculation specifics and prompts for our evaluation metrics are added in **Figure 12-15** of our revised manuscript.  We appreciate the feedback.

---

> ### Author Response · Authors · 2025-11-22
>
> > W2: The process of Logic and Evidence Extraction is not clearly described either in the main text or the appendix. While the use of GPT-4.1 is mentioned, the specific extraction procedure and output format are not illustrated clearly.
>
> Thank you for pointing out this lack of detail. You are correct that the specific procedure for Logic and Evidence Extraction needs further clarification. We describe the specific methodology as follows.
>
> Our extraction process involves three main steps, as summarized by the pseudocode below:
>
> - **Knowledge Graph Construction**: To transform unstructured text into a structured format, we first use the LLM (GPT-4.1) to iterate through each document and extract structured triples (e.g., `ExtractTriplesWithLLM`) building an initial knowledge graph (KG) for that document.
>
> - **Subgraph Processing**: Since the initial KGs can be fragmented or excessively large for efficient processing, we refine them by identifying the largest connected components. If a component remains too large, it is split into smaller, more manageable subgraphs (`SplitGraph`) to form a `SubgraphSet`.
>
> - **Logic Chain Extraction**: Finally, to capture meaningful reasoning paths, we identify high-importance 'hub' nodes within each subgraph (e.g., `SelectTopDegreeNodes`). We then perform a constrained pathfinding search (`FindAllPaths`) from these hub nodes to other nodes, and the resulting valid paths constitute our final `LogicChainSet`.
>
> Finally, both the subgraphs and the extracted logic chains are represented and stored as lists of triples.
> ```pseudo
> // Step 1: Construct Knowledge Graphs
> for each document in DocumentSet do
>     triples ← ExtractTriplesWithLLM(document)
>     KG ← BuildKnowledgeGraph(triples)
>     add KG to KnowledgeGraphSet
> end for
>
> // Step 2: Preprocess Graphs and Split Subgraphs
> for each KG in KnowledgeGraphSet do
>     subgraphs ← ExtractLargestConnectedSubgraphs(KG)
>     for each subgraph in subgraphs do
>         if NeedFurtherSplitting(subgraph) then
>             smallSubgraphs ← SplitGraph(subgraph)
>             add all smallSubgraphs to SubgraphSet
>         else
>             add subgraph to SubgraphSet
>         end if
>     end for
> end for
>
> // Step 3: Logic Chain Extraction
> for each subgraph in SubgraphSet do
>     G ← BuildDirectedGraph(subgraph)
>     sourceNodes ← SelectTopDegreeNodes(G, k)
>     for each source in sourceNodes do
>         for each target in Nodes(G) do
>             if source ≠ target then
>                 paths ← FindAllPaths(G, source, target, MinLength)
>                 add all valid paths to LogicChainSet
>             end if
>         end for
>     end for
> end for
>
> ```
> Revision：The additional detailed procedure and pseudocode are added in **Figure 11** of our revised manuscript. We appreciate the feedback.

---

### Official Review · Reviewer_dvvB · 2025-10-31

**Soundness:** 3
**Presentation:** 3
**Contribution:** 3
**Rating:** 6
**Confidence:** 4

**Summary:**

This paper systematically investigates when graph-based retrieval augmentation (GraphRAG) surpasses traditional RAG, revealing that GraphRAG excels in complex reasoning and structured knowledge integration while RAG remains more efficient for simple retrieval tasks.

**Strengths:**

This work tackles the key open question of when graph-based retrieval truly benefits RAG systems, bridging a major gap in empirical understanding. Through dense, well-controlled experiments across domains and tasks, it provides strong, data-driven evidence supporting its conclusions and design insights.

**Weaknesses:**

1. The proposed four-level task hierarchy is claimed that task difficulty increases along the retrieval difficulty and reasoning complexity. However, this paper do not provide formal or operational definitions for these levels—there are no explicit thresholds for evidence quantity, reasoning steps, or context length that determine the boundaries between levels.

2. The benchmark does not disclose the absolute corpus size for the Novel and Medical datasets. Consequently, it remains unclear how GraphRAG’s performance scales with corpus size. Without controlled experiments across small, medium, and large datasets, the current results cannot confirm whether the observed improvements stem from the graph structure itself or from differences in data volume and density.

3. The paper does not empirically analyze how different indexing contents—such as entities, relationships, phrases, or communities—affect GraphRAG performance. Although Table 15 summarizes the indexing types used by various implementations, these differences are only descriptive and intertwined with other design factors like retrieval strategies and graph density.

4. The paper does not evaluate GraphRAG on time-sensitive or temporally evolving datasets. The current study’s exclusive focus on static corpora (medical guidelines and classical novels) means it cannot assess how graph-based retrieval performs under temporal drift or continuous knowledge refresh.

**Questions:**

As discussed above in weakness.

---

> ### Author Response · Authors · 2025-11-22
>
> Dear Reviewer dvvB:
>
> We really appreciate your recognition of our work, and thanks a lot for providing insightful suggestions that can help further polish our paper.
>
> > W1: The proposed four-level task hierarchy is claimed that task difficulty increases along the retrieval difficulty and reasoning complexity. However, this paper do not provide formal or operational definitions for these levels—there are no explicit thresholds for evidence quantity, reasoning steps, or context length that determine the boundaries between levels.
>
> Thank you for pointing out the need for clearer operational definitions. In our revised paper, we will formalize the four levels based on the topological characteristics of the required knowledge subgraphs, quantified by two key metrics: Knowledge Breadth (number of unique triples) and Reasoning Depth (number of inference hops).
>
> Rather than using arbitrary rigid thresholds, we define the levels operationally by their distinct structural complexity distributions:
>
> - Fact Retrieval: Requires direct mapping between a query and a single or few connected triples.
> - Complex Reasoning: Defined by high Reasoning Depth. The task requires traversing long, multi-hop paths on the knowledge graph to deduce an answer.
> - Contextual Summarization: Defined by high Knowledge Breadth. The task requires aggregating information across a wide span of triples, though the inference chain for each piece of information may be shorter than in complex reasoning.
> - Creative Generation: The most challenging level, requiring both extreme Breadth and Depth, synthesizing dispersed information into new insights.
>
>
> Table 5: Problem Complexity Statistics of Graphrag-Bench (Novel)
> | Problem Complexity | Fact Retrieval | Complex Reasoning | Contextual Summarize | Creative Generation |
> |--------------------|----------------|--------------------|-----------------------|----------------------|
> | Knowledge Breadth  | 1.40           | 2.60               | 3.51                  | 7.11                 |
> | Reasoning Depth    | 1.69           | 6.25               | 4.64                  | 7.81                 |
>
> Table 6: Problem Complexity Statistics of Graphrag-Bench (Medical)
> | Problem Complexity | Fact Retrieval | Complex Reasoning | Contextual Summarize | Creative Generation |
> |--------------------|----------------|--------------------|-----------------------|----------------------|
> | Knowledge Breadth  | 1.25           | 3.45               | 5.10                  | 10.14                |
> | Reasoning Depth    | 1.82           | 5.23               | 4.27                  | 8.27                 |
>
> As Table 5 and 6 demonstrated, these statistics validate our operational definitions by highlighting distinct complexity dimensions rather than a simple linear progression. Specifically, we observe a structural shift between intermediate levels: while Complex Reasoning prioritizes vertical complexity with significantly deeper inference chains (Depth: 6.25) compared to Contextual Summarization (Depth: 4.64), the latter demands a wider horizontal scope of information aggregation (Breadth: 3.51 vs. 2.60). Finally, Creative Generation represents the apex of difficulty, maximizing both dimensions simultaneously to establish a clear upper bound for the hierarchy.
>
> Thank you for your feedback.

---

> ### Author Response · Authors · 2025-11-22
>
> > W2: The benchmark does not disclose the absolute corpus size for the Novel and Medical datasets. Consequently, it remains unclear how GraphRAG’s performance scales with corpus size. Without controlled experiments across small, medium, and large datasets, the current results cannot confirm whether the observed improvements stem from the graph structure itself or from differences in data volume and density.
>
> Thank you for this valuable suggestion regarding corpus size and scalability. To clarify the corpus size, we calculated token counts using `tiktoken`. The total size for the Novel dataset is approximately 1,132k tokens, and the Medical dataset is about 221k tokens. In our default experimental configuration for the Novel dataset, we construct an index for each individual book, resulting in an average effective corpus size of ~56k tokens per evaluation unit.
>
> To address the concern about how performance scales with corpus size, we conducted a controlled experiment on the Novel dataset across three scales: Small (about 56k), Medium (about 603k), and Large (about 1132k), utilizing HippoRAG2 as the GraphRAG representative. The comparative results are presented below:
>
> Table 7: The results of Generation Evaluation(ACC) of GraphRAG across novel datasets with varying corpus sizes (using HippoRAG2 as an example).
> | corpus type |      corpus size         | Fact Retrieval | Complex Reasoning | Contextual Summarize | Creative Generation|
> |-----------|---------------|-----------|-----------|-----------|-----------|
> | small   | 56k   | 60.14     | 53.38     | 64.10     | 48.28     |
> | medium   | 603k | 59.99     | 56.67     | 65.77     | 50.06     |
> | large   | 1132k | 59.19     | 54.29     | 62.63     | 51.18     |
>
> Table 8: The results of Generation Evaluation(ACC) of RAG across novel datasets with varying corpus sizes.
> | corpus type |       corpus size        | Fact Retrieval | Complex Reasoning | Contextual Summarize | Creative Generation|
> |--------------|---------------|-----------|-----------|-----------|-----------|
> | small   | 56k   | 64.73     | 58.64     | 65.75     | 60.61     |
> | medium   | 603k | 58.43     | 41.33     | 62.06     | 52.54     |
> | large   | 1132k | 58.04     | 43.20     | 62.43     | 47.19     |
>
> The comparison reveals a critical insight. Standard RAG suffers significant performance degradation as the corpus size increases. Notably, its accuracy in Complex Reasoning drops heavily from 58.64 (Small) to 43.20 (Large). **This supports the hypothesis that vector-based retrieval is prone to retrieving high-similarity but irrelevant noise as the search space expands.** In contrast, GraphRAG (HippoRAG2) exhibits remarkable stability, maintaining consistent performance across all scales (e.g., Fact Retrieval remains steady at about 59 to 60). **This confirms that the performance stability is driven by the structural constraints of the graph (i.e., explicit entity and triple matching), which effectively filter out the retrieval noise that accumulates with increasing corpus size.**
>
> Revision：The additional experiments and analysis is added in **Appendix G4** of our revised manuscript. Thank you for prompting this investigation, as it highlights a key robustness advantage of GraphRAG.

---

> > ### Author Response · Authors · 2025-11-22
> >
> > > W3: The paper does not empirically analyze how different indexing contents—such as entities, relationships, phrases, or communities—affect GraphRAG performance. Although Table 15 summarizes the indexing types used by various implementations, these differences are only descriptive and intertwined with other design factors like retrieval strategies and graph density.
> >
> > Thank you for this precise and valid observation. We agree that the specific choice of indexing content (e.g., fine-grained entities vs. coarse-grained communities) significantly impacts GraphRAG performance. Our current analysis did not provide ablations that isolate these factors, mainly because most state-of-the-art GraphRAG systems integrate indexing and retrieval so tightly that changing one component would fundamentally alter the whole pipeline. For example, some GraphRAG frameworks (e.g., HippoRAG2) refine graph connectivity beyond basic entity indexing, and this refinement is reflected in the graph’s topology.
> >
> > Given these constraints, we evaluated indexing quality by analyzing the structural properties of the constructed graphs. To make this clearer, we extended Table 5 and present additional statistics in Table 9 and 10 below. HippoRAG2 consistently exhibits substantially higher Average Degree, Clustering Coefficient, and total edges across both Novel and Medical domains. These patterns suggest that its indexing strategy captures richer and more interconnected semantic structure than earlier methods.
> >
> > Table 9: The indexing graph statistics across GraphRAG methods (Novel)
> >
> > | Metric               | MS-GraphRAG | HippoRAG2 | LightRAG | Fast-GraphRAG | HippoRAG | KET-RAG |
> > |----------------------|------------:|----------:|---------:|--------------:|---------:|--------:|
> > | **Nodes**            | 378         | 523       | 187      | 172           | 262      | 481     |
> > | **Edges**            | 272         | 2310      | 191      | 263           | 221      | 220     |
> > | **Average Degree**   | 1.48        | 8.75      | 2.10     | 3.19          | 1.73     | 1.89    |
> > | **Avg. Clust. Coeff**| 0.315       | 0.657     | 0.212    | 0.324         | 0.100    | 0.222   |
> >
> >
> > Table 10: The indexing graph statistics across GraphRAG methods (Medical)
> >
> > | Metric               | MS-GraphRAG | HippoRAG2 | LightRAG | Fast-GraphRAG | HippoRAG | KET-RAG |
> > |----------------------|------------:|----------:|---------:|--------------:|---------:|--------:|
> > | **Nodes**            | 182         | 598       | 115      | 127           | 207      | 386     |
> > | **Edges**            | 166         | 3979      | 149      | 350           | 213      | 321     |
> > | **Average Degree**   | 1.82        | 13.31     | 2.58     | 5.50          | 2.06     | 2.41    |
> > | **Avg. Clust. Coeff**| 0.300       | 0.497     | 0.139    | 0.347         | 0.087    | 0.304   |
> >
> >
> > These structural metrics act as a practical indicator of how much useful information each indexing strategy captures. Of course, evaluating different indexing contents would offer a more fine-grained assessment of the graph structures built by each GraphRAG framework and more clearly reveal their respective strengths. We will explore how to design evaluations that target different graph elements in future work. We sincerely thank you again for this valuable suggestion.
> >
> > > W4: The paper does not evaluate GraphRAG on time-sensitive or temporally evolving datasets. The current study’s exclusive focus on static corpora (medical guidelines and classical novels) means it cannot assess how graph-based retrieval performs under temporal drift or continuous knowledge refresh.
> >
> > Thank you for this highly insightful suggestion. Evaluating RAG performance on temporally evolving datasets is indeed a valuable research direction. As you noted, existing work (such as StreamingQA[1] and TimeQA[2]) highlights the importance of assessing how systems handle knowledge from different time spans or handle shifts in information over time.
> >
> > We honestly acknowledge that the current version of GraphRAG-Bench is built upon static corpora (novels and specific medical guidelines). However, your point provides a key direction for our future work, and our existing data is well-suited for such an extension. Specifically, our medical dataset uses the NCCN Guidelines, which are updated annually. These versioned updates provide an ideal foundation for a dynamic knowledge refresh benchmark, as they capture concrete changes like new treatment protocols and diagnostic criteria. In our future work, we plan to investigate how to incorporate these updates, enabling GraphRAG-Bench to evaluate performance on both static knowledge bases and dynamic, evolving versions. We are grateful for this suggestion, as it offers significant guidance for the future optimization of our work.
> >
> > [1] StreamingQA: A Benchmark for Adaptation to New Knowledge over Time in Question Answering Models
> >
> > [2] A Dataset for Answering Time-Sensitive Questions

---

> > > ### Comment · Reviewer_dvvB · 2025-11-27
> > > **Response to the Rebuttal**
> > >
> > > I appreciate the authors' response to my questions. I tend to maintain the current positive score.

---

> ### Author Response · Authors · 2025-11-27
>
> Dear Reviewer dvvB,
>
> Thank you for your expertise and recognition. We appreciate your insightful comments, which have guided us in improving the manuscript. Following your suggestions, we have made the corresponding revisions in the updated version.
>
> Best regards,
> Authors of Paper15574

---

### Official Review · Reviewer_zK5v · 2025-10-31

**Soundness:** 4
**Presentation:** 4
**Contribution:** 3
**Rating:** 6
**Confidence:** 5

**Summary:**

This paper discusses and tests the advantages and disadvantages of GraphRAG compared to vanilla RAG. To achieve this goal, the authors discusses the limitations of existing benchmarking datasets for GraphRAG/RAG-related systems and propose GraphRAG-Bench, a benchmark on knowledge retrieval and contextual reasoning. GraphRAG-Bench not only focus on testing a model’s ability to correctly retrieve necessary information, but also tests the model’s ability to reason in complex scenarios. It contains 4 types of tasks: Fact Retrieval, Contextual Summarize, Complex Reasoning and Creative Generation. The process of constructing the dataset is also introduced, with 6-steps - Corpus collection, Logical Mining, Evidence Collection, Question Generation, Check, and Refinement. With this benchmark, authors systematically checked the condition when a series of GraphRAG works.

**Strengths:**

1.	This paper studied a fundamental and important problem. GraphRAG is trending, demonstrating good motivation. The proposed benchmark provides rooms to systematically examine the advantage of different GraphRAG systems.

2.	This manuscript identifies key limitations of existing RAG benchmarking datasets, including neglecting the evaluation of logical reasoning, limited corpus coverage, and focusing only on end results.

3.	Apart from general QA, the proposed benchmark includes a set of tasks, including Fact Retrieval, Contextual Summarize, Complex Reasoning and Creative Generation, which allows users to comprehensively evaluate retrieval and reasoning capabilities of RAG systems from different perspectives.

**Weaknesses:**

1.	The dependence of GraphRAG on LLM ability. The authors tested two models (GPT-4o-mini and Qwen2.5-14B) on the benchmark. However, the analysis of how GraphRAG depends on the ability (size) of LLMs is missing. It would be nice if the author could give some analysis on the minimum size for a successful GraphRAG.

2.	Some of the multi-hop QA datasets, which contains QA pairs with various question types and different difficulties, such as CWQ, MuSiQue, and 2WikiMultihopQA, are not discussed in this manuscript.

3.	This paper lacks a concrete explanation of how the raw text corpus is organized into an ontology by GPT-4.1. This task can be challenging, hence one may need to recognize entities and relations, and then perform necessary disambiguation procedures.

4.	The proposed benchmark does not evaluate the efficiency of graph construction. Ignoring graph construction efficiency is justifiable only if the constructed graph can be reused across a wide range of scenarios.

**Questions:**

1.	This reviewer is interested to know the reason why some GraphRAG systems, such as ToG, ToG2.0, GNN-RAG, SubgraphRAG are not taken into comparison.

---

> ### Author Response · Authors · 2025-11-22
>
> Dear Reviewer zK5v,
>
> Thanks a lot for your detailed feedback. We really appreciate your recognition of our work and also appreciate your time and effort in providing insightful suggestions that can help further polish our paper. Below are detailed responses to your comments and suggestions:
>
> > W1: The dependence of GraphRAG on LLM ability. The authors tested two models (GPT-4o-mini and Qwen2.5-14B) on the benchmark. However, the analysis of how GraphRAG depends on the ability (size) of LLMs is missing. It would be nice if the author could give some analysis on the minimum size for a successful GraphRAG.
>
> Thank you for this valuable suggestion. To explore how GraphRAG depends on LLM size, we conducted additional experiments comparing Standard RAG and GraphRAG (using HippoRAG2 as the representative) across the Qwen2.5 series (3B, 7B, and 14B) on the Medical dataset.
>
> Since GraphRAG constructs graph structures offline during the indexing phase, these indices can be reused. Therefore, we focused our investigation on the impact of model size during the online inference phase. Specifically, for the GraphRAG experiments, we utilized the same pre-constructed index (built with GPT-4o-mini) used in our main experiments. This allows us to isolate the impact of the model's size during the generation phase. The results are presented below:
>
>
> Table 1: GraphRAG (HippoRAG2) performance (ACC) on medical dataset across Qwen series models (3B–14B)
> | Model       | Fact Retrieval | Complex Reasoning | Contextual Summarize | Creative Generation | Avg |
> |-------------|-----------|-----------|-----------|-----------|---------|
> | Qwen2.5-3b  | 60.19     | 58.11     | 58.69     | 51.34     | 57.08   |
> | Qwen2.5-7b  | 65.65     | 64.25     | 64.26     | 51.85     | 61.50   |
> | Qwen2.5-14b | 65.98     | 62.62     | 64.95     | 62.09     | 63.91   |
>
>
> Table 2: RAG performance(ACC) on medical dataset across Qwen series models (3B–14B)
>
> | Model       | Fact Retrieval | Complex Reasoning | Contextual Summarize | Creative Generation | Avg |
> |-------------|-----------|-----------|-----------|-----------|---------|
> | Qwen2.5-3b  | 58.66     | 52.25     | 58.71     | 54.89     | 56.13   |
> | Qwen2.5-7b  | 57.45     | 53.53     | 60.52     | 55.65     | 56.79   |
> | Qwen2.5-14b | 57.56     | 56.01     | 61.95     | 60.91     | 59.10   |
>
> As Table 1 and 2 illustrated, GraphRAG exhibits a higher sensitivity to model capacity than Standard RAG. While RAG's performance remains relatively flat across scales (Avg 56.13 → 59.10), GraphRAG demonstrates substantial growth (Avg 57.08 → 63.91), indicating a greater reliance on the model's reasoning ability to synthesize structural information. **Notably, we observe a distinct performance inflection point at the 7B parameter scale (Avg increasing from 57.59 to 61.50), suggesting that 7B serves as a practical "minimum size" threshold where the model acquires sufficient reasoning power to effectively leverage graph-based context.**
>
> Revision：The additional experiments and analysis are added in **Appendix G.4** of our revised manuscript.
> We truly appreciate your valuable suggestions.

---

> ### Author Response · Authors · 2025-11-22
>
> > W2: Some of the multi-hop QA datasets, which contains QA pairs with various question types and different difficulties, such as CWQ, MuSiQue, and 2WikiMultihopQA, are not discussed in this manuscript.
>
> We sincerely thanks for your expertise and insightful feedback. You are correct that these are typical representatives of multi-hop datasets. We would like to clarify the reason why we choose MultiHop-RAG and HotpotQA datasets to compare with GraphRAG-Bench. Because we found that exist multi-hop QA datasets have some same following questions:
>
> First, regarding the question type, while datasets like MuSiQue and 2WikiMultihopQA have diverse multi-hop questions, we find they share a key limitation with HotpotQA: they primarily test "retrieval difficulty" (locating scattered facts) rather than the "reasoning difficulty" (synthesizing interconnected concepts). We provide a detailed analysis of this limitation in Section 2.2 (Current RAG benchmarks) of our paper. Furthermore, GraphRAG-Bench is designed to evaluate a wider range of scenarios not only multi-hop Complex Reasoning, but also have Fact Retrieval and Contextual Summarize, to provide a more comprehensive assessment.
>
> Second, the corpus structures in these benchmarks are typically created by merely aggregating the context of each question. Therefore, knowledge points are scattered across disparate contexts with few meaningful connections. This leads to corpora with low information density and weak inter-entity connectivity, lacking the rich and hierarchical dependencies necessary to comprehensively assess GraphRAG’s strengths.
>
> To further enrich our analysis, we expanded our corpus statistical analysis (shown in Table 13) to include MuSiQue and 2WikiMultihopQA. (Due to time limits and the lack of a available corpus for CWQ, we didn't include it in this analysis). The result is shown in Table 3.
>
> Table 3: Corpus statistics for different benchmarks.
> | **Benchmark** | **Avg. Entities** | **Avg. Relations** | **Prop. of Non-isolated Entities** | **Avg. Degree** | **Prop. Degree > 1** | **Prop. Degree > 2** | **Prop. Degree > 3** | **Avg. Component Size** |
> |:--------------|:-----------------:|:------------------:|:----------------------------------:|:----------------:|:--------------------:|:--------------------:|:--------------------:|:-----------------------:|
> | MuSiQue(additional) | 21.5 | 6.5 | 0.39 | 0.60 | 0.23 | 0.10 | 0.06 | 2.33 |
> | 2WikiMultihopQA(additional) | 41.9 | 13.4 | 0.40 | 0.64 | 0.23 | 0.11 | 0.07 | 2.09 |
> | UltraDomain | 170.6 | 73.2 | 0.40 | 0.86 | 0.27 | 0.15 | 0.09 | 2.71 |
> | MultiHop-RAG | 10.1 | 3.82 | 0.41 | 0.76 | 0.26 | 0.14 | 0.09 | 2.70 |
> | HotpotQA | 39.3 | 12.7 | 0.41 | 0.65 | 0.23 | 0.12 | 0.06 | 2.11 |
> | **GraphRAG-Bench (novel)** | 19.6 | 20.9 | 0.66 | 2.27 | 0.47 | 0.25 | 0.17 | 3.99 |
> | **GraphRAG-Bench (medical)** | 11.8 | 6.2 | 0.48 | 1.05 | 0.36 | 0.20 | 0.12 | 3.15 |
>
> As the Table 3 demonstrates, MuSiQue and 2WikiMultihopQA are statistically similar to HotpotQA, featuring sparse graphs with a low Average Degree (about 0.6) and a low Proportion of Non-isolated Entities (about 0.4). This indicates a high degree of fragmentation. In contrast, both datasets of GraphRAG-Bench(novel and medical) are significantly denser. Novel dataset shows a much higher Average Degree (2.27) and Prop. of Non-isolated Entities (0.66). Medical dataset also clearly surpasses the baselines in key connectivity metrics (Avg. Degree 1.05, Avg. Component Size 3.15). This confirms our benchmark provides the complex relational structure essential for evaluating GraphRAG's reasoning capabilities, which is lacking in the other datasets.
>
> Revision：The additional corpus statistics is added in **Table 14** of our revised manuscript. We appreciate your suggestion and consideration.

---

> ### Author Response · Authors · 2025-11-22
>
> > W3: This paper lacks a concrete explanation of how the raw text corpus is organized into an ontology by GPT-4.1. This task can be challenging, hence one may need to recognize entities and relations, and then perform necessary disambiguation procedures.
>
> Thank you for pointing out this lack of detail. Organizing raw text into a structured ontology involves multiple steps. We describe the specific details as follows:
>
> Our ontology construction process involves four main steps, as summarized by the pseudocode below.
>
> - **Text Extraction**: To extract plain text from the raw NCCN guidelines (in PDF format), we utilize a PDF parsing tool to convert the documents into a machine-readable format.
> - **Instance-Level Graph Construction**: Since the raw text is dense and contains significant context irrelevant to specific cases, we need to isolate potential relations for each disease. Therefore, for each specific disease section, we use GPT-4.1 to identify entities and relations, extracting structured triples to build an initial instance-level Knowledge Graph (KG).
> - **Ontology Induction**: To organize this information into a structured format, we input both the extracted triples and their corresponding original text chunks into the LLM to induce a high-level ontology schema (classes and hierarchy).
> - **Refinement**: Finally, we refine the generated ontology by cross-referencing it with the original text to resolve ambiguities and ensure the schema accurately reflects the domain structure.
>
> ```psude
> // Step 1: Text Extraction
> guidelines ← ParsePDF(NCCN_Guidelines)
>
> // Step 2: Instance-Level KG Construction
> for each disease_section in guidelines do
>     // Extract entities and relations to form triples
>     triples ← ExtractTriplesWithLLM(disease_section)
>     KG ← BuildInstanceGraph(triples)
>     add KG to InstanceKGSet
> end for
>
> // Step 3: Ontology Induction
> for each KG in InstanceKGSet do
>     // Induce abstract schema (Ontology) from instance triples by GPT-4.1
>     rawOntology ← GenerateOntologyFromTriples(KG.triples, contexts)
>     add rawOntology to RawOntologySet
> end for
>
> // Step 4: Ontology Refinement
> for each ontology in RawOntologySet do
>     originalText ← GetSourceText(ontology)
>     // Refine structure and resolve ambiguity using source context
>     refinedOntology ← RefineAndDisambiguate(ontology, originalText)
>     add refinedOntology to FinalOntologySet
> end for
> ```
>
> Revision：The additional corpus statistics is added in **Figure 11** of our revised manuscript. We appreciate the feedback.

---

> ### Author Response · Authors · 2025-11-22
>
> > W4: The proposed benchmark does not evaluate the efficiency of graph construction. Ignoring graph construction efficiency is justifiable only if the constructed graph can be reused across a wide range of scenarios.
>
> Thank you for this valuable suggestion. Evaluating graph construction efficiency provides a useful reference for real-world development and highlights the practical efficiency of different GraphRAG methods. We have conducted additional experiments to address this.
>
> We selected the Novel dataset for this evaluation, using one book (approximately 56k tokens) for graph construction. We recorded the construction time, total input tokens (per 1k corpus tokens), and total output tokens (per 1k corpus tokens). The results are as follows:
>
> Table 4: Construction time and average token usage (per 1k corpus tokens) during the indexing phase across different GraphRAG methods.
> | **Method**        | **Time (s)** | **Input Tokens** | **Completion Tokens** | **Total Tokens** |
> |-------------------|--------------|-----------------|---------------------|-----------------|
> | MS-GraphRAG       | 292.45       | 9568.4          | 2122.2              | 11690.6         |
> | LightRAG          | 710.32       | 7203.6          | 1263.8              | 8467.4          |
> | Fast-GraphRAG     | 281.74       | 3352.9          | 1143.8              | 4496.7          |
> | HippoRAG          |  77.42       | 2767.6          | 410.2               | 3177.8          |
> | HippoRAG2         |  96.71       | 5062.4          | 830.2               | 5892.6          |
> | RAPTOR            | 135.21       | 2029.3          | 33.9                | 2063.2          |
> | KGP               |  32.01       | 1223.9          | 369.2               | 1593.1          |
> | KET-RAG           | 350.43       | 7735.6          | 1506.4              | 9242.0          |
> | LAZY-GraphRAG     | 253.59       | 9274.9          | 1281.4              | 10556.3         |
>
>
> We summarize the observations as follows. The efficiency varies significantly across different GraphRAG methods. Notably, KGP demonstrates the highest efficiency, completing the construction process in just 32.01 seconds with the lowest average token usage (1593.1). HippoRAG and HippoRAG2 also perform efficiently, maintaining low time and token costs. In contrast, LightRAG and the MS-GraphRAG incur much higher resource consumption; LightRAG requires over 710 seconds to finish, while MS-GraphRAG consumes nearly 11690.6 tokens. These results suggest that lightweight frameworks offer a far more cost-effective solution for graph indexing, which is critical for applications with limited time or computational budgets.
>
> Revision：The additional experiments and analysis is added in **Appendix G3** of our revised manuscript. We deeply appreciate your valuable advice.
>
>
> > Q1: This reviewer is interested to know the reason why some GraphRAG systems, such as ToG, ToG2.0, GNN-RAG, SubgraphRAG are not taken into comparison.
>
> Thank you for this thoughtful question. ToG, ToG2.0, GNN-RAG, SubgraphRAG are indeed important and effective frameworks. But we think they are actually not a typical GraphRAG frameworks, we categorize them as contributions to Knowledge Graph Question Answering (KGQA). Therefore we don't add these baselines. The detail reason are following:
>
> The primary distinction is that the KGQA setting assume an existing, structured knowledge graph is available, and its methods are designed for retrieval and reasoning on that graph. However, in real-world scenarios, many domain texts are unstructured and lack a pre-existing, well-constructed knowledge graph. Therefore, GraphRAG must process this raw text: it is required to first construct a knowledge graph during the indexing phase and then perform retrieval on this self-constructed graph to generate an answer. This is the key difference from KGQA.
>
> Because our study focuses on the performance of the entire GraphRAG pipeline (i.e., graph construction, retrieval, and generation), we centered our comparison on methods designed for this end-to-end task. We believe this focus allows for a clearer and more direct evaluation relevant to our paper's specific motivation.

---

> > ### Comment · Reviewer_zK5v · 2025-11-24
> > **Response to the Rebuttal**
> >
> > This reviewer appreciates the authors’ efforts to clarify the weaknesses and address the questions raised in the review.
> >
> > Regarding W2, this reviewer understands that, compared with 2Wiki, HotpotQA and MuSiQue have a certain degree of similarity in nature. Nevertheless, 2wiki and MuSiQue cover 3-5 hop questions that are more difficult compared to the 2-hop questions in HotpotQA. The differences in difficulty motivate the reviewer to raise the weakness.
> >
> > Regarding W3, this reviewer suggests the authors include some prompts and scripts for ontology construction.
> >
> > Regarding the reply to Q1, this reviewer appreciates the clarification, and would like to encourage the authors to include the clarification in the manuscript.
> >
> > Given the above reasons, this reviewer supports the acceptance of the paper, and would like to maintain the positive rating.

---

> > > ### Author Response · Authors · 2025-11-25
> > >
> > > Dear Reviewer zK5v,
> > >
> > > Thank you for your recognition of our work and for providing such thorough and insightful feedback. Your comments and suggestions are invaluable in helping us improve the quality and clarity of our work. Follwing your suggestions, we have added the additional experiments and analysis in the revised manuscript.
> > >
> > > Best regards
> > >
> > > Authors of Paper15574

---

### Author Response · Authors · 2025-12-02
**General Response**

Dear Reviewers and Area Chairs:

We sincerely thank you for your time, effort, and invaluable feedback during the author-reviewer discussion phase. Your insights have been crucial in helping us refine and improve our work. Below, we would like to briefly summarize the major contribution of our paper and the key points raised during the rebuttal stage.

## **Overall Assessment**
**We received the initial scores of** $\textbf{\textcolor{maroon}{8,6,6,6}}$. And these scores **remained consistently positive** during the rebuttal stage. Prior to the API incident, Reviewers zK5v and dvvB explicitly stated that they would maintain the positive scores, expressing recognition for our work and rebuttal in their comments.

> Reviewer zK5v: Given the above reasons, I support the acceptance of the paper, and would like to maintain the positive rating.

> Reviewer dvvB:  I tend to maintain the current positive score.


## **Strengths of Our Work**
- **Clear motivation**: This work addresses the fundamental question of when Graphs should be used in RAG systems, filling an important gap in current empirical understanding (Reviewer zK5v, dvvB, bvTB).
- **Well-designed benchmarks**: It introduces a comprehensive benchmark that fixes limitations in prior work by adding multi-level task structures and clear quantitative metrics that better reflect real-world complexity (Reviewer zK5v, ebWh).
- **Solid experiments**: It presents extensive, carefully controlled experiments across diverse tasks, methods, and LLMs, providing solid experiments to support its conclusions (Reviewer dvvB, ebWh, bvTB).
- **Practical and actionable guidance**: It offers concrete guidelines and design principles that can directly guide future GraphRAG system development (Reviewer ebWh, bvTB).

## **Addressed Concerns**
- **The impact of model size**: Reviewer zK5v suggested us to evaluate "how GraphRAG depends on the ability (size)" of LLMs, we conducted comparative experiments using Qwen2.5-series(3B, 7B and 14B) for both RAG and GraphRAG (represented by HippoRAG2). The results are presented in Tables 1-2 (added to Appendix G.4).
- **The impact of dataset scale**: Reviewer dvvB concerns about the lack of "controlled experiments across small, medium, and large datasets," we categorized the Novel dataset into small (56k), medium (603k), and large (1,132k) scales based on token counts. We evaluated the performance differences of RAG and GraphRAG (HippoRAG2) across these scales. The results are shown in Tables 7-8 (added to Appendix G.4).
- **Analysis on graph construction efficiency**: Reviewer zK5v suggested adding an analysis of "graph construction efficiency." We recorded the construction time, input tokens, and output tokens consumed by the LLM for various GraphRAG methods. The results are presented in Table 4 (added to Appendix G.3).
- **Details on benchmark construction and evaluation**: Reviewer zK5v, ebWh, and bvTB noted the absence of specific prompts and implementation details regarding benchmark construction and evaluation. We have added detailed explanations of the logic chain construction for the Novel dataset and ontology construction for the Medical dataset. We also included the exact prompts for specific evaluation, such as EVIDENCE RECALL, FAITHFULNESS, and Evidence Coverage. These details have been illustrated in Figures 12-15 in the revised manuscript.

Thank you again for your recognition of our work and for providing such thorough and insightful feedback. Your comments and suggestions are invaluable in helping us improve the quality and clarity of our work. Following your suggestions, we have added the additional experiments and analysis in the revised manuscript.

Hope that this summary could facilitate the next stage of review and discussion.

Best regards,

Authors of Paper15574

---

### Meta-Review · Area_Chair_Q4mY · 2026-01-08

**Summary:**

This paper addresses the gap in empirically evaluating Graph RAG systems by introducing GraphRAG-Bench. The benchmark features two distinct corpora (a Medical dataset and a loose, unstructured Novel dataset) and organizes evaluation around a four-level task hierarchy: Fact Retrieval, Complex Reasoning, Contextual Summarization, and Creative Generation. The authors propose stage-specific metrics to assess the entire pipeline, including graph quality, retrieval relevance, and generation faithfulness. Key insights demonstrate that while GraphRAG incurs higher preprocessing costs, it outperforms vanilla RAG on complex reasoning and summarization tasks and exhibits superior robustness as corpus size increases.

**Reviewer Concerns:**

The reviewers were generally positive about the motivation and the comprehensiveness of the benchmark. Several concerns were raised and subsequently addressed during the rebuttal phase:

Dependence on LLM Capability: Reviewer zK5v questioned the minimum model size required for effective GraphRAG. The authors addressed this by adding experiments with the Qwen2.5 series (3B, 7B, 14B), identifying 7B as a practical performance inflection point.

Scalability with Corpus Size: Reviewer dvvB noted a lack of analysis regarding corpus size. The authors conducted controlled experiments across different scales, demonstrating that vanilla RAG performance degrades with scale while GraphRAG remains stable.

Graph Construction Efficiency: Reviewer zK5v requested analysis on construction costs. The authors added a detailed breakdown of time and token usage for various methods, highlighting the cost-performance trade-offs.

Methodological Clarity: Reviewers ebWh and bvTB flagged unclear definitions for metrics and dataset construction. The authors provided detailed pseudocode and prompts for these processes in the revised appendix.

Task Definitions: Reviewer dvvB asked for operational definitions of task difficulty. The authors clarified this by quantifying "Knowledge Breadth" and "Reasoning Depth" for each level.

**Reviewer Scores:**

no changes

---

### Decision · Program_Chairs · 2026-01-26

Accept (Poster)